# Video inoculation against election misinformation across 12 EU nations
Mikey Biddlestone [1,2] ✉, Beth Goldberg [3], Melisa Basol[2], Katie Washington[2], Sara Elnusairi[3], Anneka Sharpley[3], Meghan Graham[4], Sander van der Linden [5], Ricky Green[1], Jon Roozenbeek[5], Rachel Xu [3] & Andrew Pel [6]

Short video-based prebunking reduces misinformation susceptibility by forewarning viewers about manipulation tactics. However, its effectiveness across older populations (45 + ), diverse cultures, and election-related misinformation remains unclear. To address this, we conducted 13 surveys across 12 EU nations ($N = 19,735$), testing three inoculation videos developed for the largest prebunking campaign to date, which reached 120M+ YouTube users before the 2024 EU Elections. These videos targeted three widely used misinformation tactics—scapegoating, decontextualization, and discrediting—which were prevalent across EU nations leading up to the elections. The videos improved manipulation (measured through manipulativeness assessments) and technique (measured through the correct identification of tactics) discernment of manipulative content from non-manipulative content and enhanced sharing decisions. Though effects were small ($d$s ≈ 0.08–0.38), they were significant across surveys. Longer (50s) videos showed more consistent improvements in discernment than shorter (20s) ones, but both improved technique discernment. Moderation analyses revealed country-level (e.g., education index) and socio-demographic (e.g., personal educational attainment) influences that could inform future interventions. These findings demonstrate that scalable video-based inoculation interventions can be deployed to counter election misinformation across nations, but future work should explore how repeated or context-specific prebunking can sustain resistance to misinformation in diverse electoral and media environments.

A Eurobarometer poll found that 82% of Europeans agree that false or misleading news is a problem for democratic functioning[1]. Such threats are particularly salient in the context of EU elections, where both foreign and domestic actors have sought to amplify doubts about the fairness of electoral processes[2,3]. Beliefs in election manipulation can erode public confidence and contribute to polarization, making it essential to develop interventions that help citizens evaluate information reliability independently, without relying on central "arbiters of truth"[4].

One well-known method of building resilience against not only false claims but also the tactics that underlie mis- and disinformation is *prebunking*, or pre-emptive debunking. The use of prebunking in contemporary interventions is informed by inoculation theory[5]. Inoculation interventions comprise two main synergistic components: (1) a forewarning of possible attempts at (malicious) persuasion, and (2) pre-emptive refutations of how this persuasion might occur (with defanged examples, or *microdoses*). Research has demonstrated this strategy to be effective at

reducing misinformation susceptibility—that is, in the perceived reliability of false or misleading information, regardless of the intention to deceive[6]. For example, a recent meta-analysis by Lu et al.[7] found that inoculation interventions had a modest but significant effect on improving veracity discernment, a measure of distinguishing reliable (or true) from unreliable (or false) information ($d = 0.20$; see also Simchon et al.[8]).

Inoculation interventions usually come in one of three forms[9]: games[10,11], short pieces of text or infographics[12], or short (+- 1 min. long) videos[13,14]. While some research is available on the cross-cultural efficacy of inoculation interventions[11,15,16], inoculation interventions are yet to be tested across a wider variety of cultural contexts, in samples with an average age above 40—a sample demographic known to predict higher likelihood of sharing misinformation online[17] (although other work indicates that this can be explained by lower verbal IQ and numeracy skills in both younger and older adults[18])—and targeting prevalent manipulative narratives during significant societal events (e.g., elections). To address these gaps, we present

[1]School of Psychology, University of Kent, Canterbury, UK. [2]Moonshot, London, UK. [3]Google Jigsaw LLC, New York, NY, USA. [4]Independent, New York, NY, USA. [5]Department of Psychology, University of Cambridge, Cambridge, UK. [6]Moonshot, Toronto, ON, Canada. ✉e-mail: mikeybiddlestone@gmail.com

the results of a study into the efficacy of scalable inoculation videos that preemptively refute three manipulation tactics that are prevalent during election campaigns: scapegoating individuals or groups, decontextualizing information by leaving out essential context, and discrediting opponents through ad hominem attacks[13]. Our findings come from a large sample of participants across 12 EU nations aged 45 and older.

Tackling mis- and disinformation in a rights-preserving manner is an urgent societal challenge[19]. To address this, Moonshot (a research-based non-profit tackling online harms), Jigsaw (a unit within Google), and Google developed inoculation videos targeting three manipulation tactics commonly seen during election cycles. The first video covered scapegoating, highlighting how individuals or groups are unfairly blamed for complex issues they are not solely responsible for[13,20]. The second addressed decontextualization, showing how information can be distorted when removed from its original context to present content in a biased or manipulative way[16]. The third tackled discrediting, exposing how personal attacks are used to undermine credibility[21]. During a paid media campaign for the June 2024 EU elections, these videos were distributed across YouTube and Meta in Germany, Belgium, France, Italy, and Poland. While Google measured effectiveness via YouTube brand lift surveys (average fieldwork improvement = 1.53%; translates to ~9.18% improvement in lab settings[13,22]), it was not possible to evaluate the effectiveness of the other videos that were created for other European languages, which also had native-language voiceovers. We therefore tested their impact on manipulation discernment in controlled lab experiments across 12 nations, 3 months after the elections.

Building on previous work[13,14], we examined the replicability and conditional effects of video-based inoculation interventions. Specifically, we tested their effectiveness across cultures, among older populations, and explored factors that may influence their impact. Most research on inoculation theory has focused on US samples, with some studies conducted in a few European nations (e.g., Germany, France[23]) and far fewer in non-Western contexts (e.g., China, Nigeria[7,16,24]). While Iyengar et al.[25] found significant improvements in discernment in an Indian sample, Harjani et al.[15] reported null effects in another Indian sample. To better understand cross-cultural variation, we tested the effectiveness of our inoculation videos across 12 EU nations that held elections in June 2024.

Older populations are particularly vulnerable to political manipulation due to digital education exclusion[26] and their heavy reliance on familiar sources such as news portals and social media[27,28]. They are also more likely to share misinformation[17] and, given their higher voter turnout[29], represent a key consequential demographic for political inoculation efforts. Brashier and Schacter[30] identify three factors that may influence misinformation susceptibility across age groups: cognitive decline, social change, and digital literacy. Cognitive decline can increase misinformation susceptibility by impairing memory for information sources and reinforcing reliance on familiar, repeated claims[31]. Social change may also play a role, as greater interpersonal trust (or reduced institutional trust) over the lifespan may make individuals more likely to believe misinformation from close or trusted sources, or less from official or governmental sources. Additionally, the declining cognitive component of digital literacy—critical thinking[32]—over the lifespan is linked to weaker misinformation resistance[33,34]. Finally, older adults are rarely in education, unlike people under 20, which complicates the implementation of educational or literacy programs among this demographic. To promote digital inclusion among this important demographic and replicate previous inoculation effects among younger age groups, we tested our inoculation videos among participants aged 45 and older.

To understand whether the prebunking interventions demonstrated broad efficacy across populations, or whether there are certain populations and factors that require attention to improve receptivity to these interventions, we included several moderator variables. To examine how country-level factors might influence the effectiveness of our inoculation videos across 12 nations, we included several moderators: Western vs. non-Western Europe (measured by longitude), national education indices, industrialization indices, GDP per capita, and democratic indices. These

moderators were included to provide insights into the cross-cultural replicability of prebunking interventions. For example, while some evidence indicates cross-cultural efficacy of prebunking games[12], other evidence suggests the replicability of these findings could be limited[15,16]. To assess age-related effects, we included the moderators of digital literacy[33] and political intolerance[35]. These were included to assess the replicability of prebunking interventions across individual differences that vary as we age (i.e., digital literacy is lower among older adults[33]) and based on temporally relevant changes (i.e., political tolerance has increased over time despite age differences[36]). Additionally, we accounted for misinformation perceptions (general manipulation discernment ability) and socio-demographic factors (age, gender, education, political ideology) to comprehensively explore the potential individual difference moderators related to existing misinformation skills and societal cohorts of inoculation interventions.

To understand how individuals engage with manipulative content online, we examine four key outcome variables that together capture different aspects of misinformation discernment and susceptibility. First, technique discernment refers to individuals' abilities to correctly identify the manipulative techniques (e.g., scapegoating) embedded in content[13,37]. This measure taps into cognitive awareness of the categorization of manipulation tactics and reflects an important component of media literacy—the skill of detecting influence strategies designed to mislead or persuade. Second, manipulation discernment captures participants' ability to judge the relative manipulativeness of content, as indicated by higher ratings of manipulative items compared to non-manipulative ones[13]. This measure reflects a more intuitive or implicit evaluation of content credibility. Third, we assess sharing intentions, operationalized as participants' stated willingness to share manipulative versus non-manipulative content. Sharing intentions are a critical behavioral proxy for misinformation spread, directly relevant to real-world consequences of exposure[38], and even indicative of real-world sharing habits[39–41]. Finally, confidence in manipulation detection measures how sure participants are in their ability to tell when content is manipulative. This metacognitive component provides insight into individuals' self-perceived epistemic vigilance—important because overconfidence or underconfidence may influence whether people act on their judgments (e.g., by sharing or fact-checking). Together, these dependent variables offer a comprehensive picture of how individuals process, evaluate, and respond to manipulative content.

## Overview of the current research

To test the effectiveness of the three prebunking videos, we ran 13 identical surveys across 12 EU nations: Belgium (in both French and Flemish), Bulgaria, France, Germany, Hungary, Ireland, Italy, Lithuania, the Netherlands, Poland, Romania, and Spain. The surveys in France, Germany, and Italy also included shorter versions of the prebunking videos for comparison. In each survey, participants were randomly allocated to watch one of the videos (or a control video of Lionel Messi's football career highlights) and were then asked to indicate their responses in an item rating task of manipulative and non-manipulative content. This task was used to measure their manipulation discernment through manipulativeness assessments, technique discernment through the correct detection of manipulation tactics vs. non-manipulative content, sharing decisions through willingness to share manipulative vs. non-manipulative content, and their confidence in detecting manipulation (see "Methods" section for full details).

## Pre-registered directional hypotheses

H1) Watching a relevant prebunking video will significantly increase manipulation discernment (i.e., rating manipulative content as more manipulative and non-manipulative content as less manipulative compared to the control group) for that tactic.

H2) Watching a relevant prebunking video will significantly increase technique discernment (i.e., correctly identifying the relevant manipulation tactic for manipulative items and "None of the above" for non-manipulative items more frequently than the control group) for that tactic.

H3) Watching a relevant prebunking video will significantly increase manipulativeness assessments (i.e., higher manipulativeness ratings for manipulative items compared to the control group) for that tactic.

H4) Watching a relevant prebunking video will significantly increase technique recognition of manipulation (i.e., correctly identifying the relevant manipulation tactic for manipulative items more frequently than the control group) for that tactic.

H5) Watching a relevant prebunking video will significantly improve sharing decisions (i.e., lower willingness to share manipulative items and higher willingness to share non-manipulative items compared to the control group) for that tactic.

H6) Watching a relevant prebunking video will significantly decrease willingness to share manipulative content compared to the control group for that tactic.

## Pre-reigistered exploratory research questions

RQ1) Will watching a relevant prebunking video significantly affect *manipulativeness assessments of non-manipulative content* (i.e., lower manipulativeness ratings for non-manipulative items compared to the control group) for that tactic?

RQ2) Will watching a relevant prebunking video significantly affect *technique recognition of non-manipulative content* (i.e., more frequent correct "None of the above" responses for non-manipulative items than the control group) for that tactic?

RQ3) Will watching a relevant prebunking video significantly affect *willingness to share non-manipulative content* (i.e., higher willingness to share non-manipulative items compared to the control group) for that tactic?

RQ4) Will watching a relevant prebunking video significantly affect *confidence in detecting manipulation* compared to the control group for that tactic?

RQ5) *Cross-protection*: Will watching a prebunking video for one tactic improve manipulation discernment, technique discernment, and sharing decisions compared to the control group for other, non-prebunked tactics? Results for these analyses can be found in the Supplement (Section 1).

RQ6) *Skepticism/distrust*: Will any prebunking condition increase skepticism (i.e., rating both manipulative and non-manipulative content as more manipulative, or selecting a manipulation tactic for both manipulative and non-manipualtive items more frequently than the control group)? Results for these analyses can be found in the Supplement (Section 9).

RQ7) *Naïveté/gullibility*: Will any prebunking condition increase naïveté (i.e., rating both manipulative and non-manipulative content as less manipulative, or selecting "None of the above" for both manipulative and non-manipulative items than the control group)? The results of these analyses can be found in the Supplement (Section 9).

RQ8) Will any hypothesized or exploratory effects be moderated by any of the following variables: general manipulation discernment, voting behavior in the June EU elections, intentions to share videos within one's social network, digital literacy, socio-demographics (age, gder, education, political ideology), or country-level indices (longitude, education index, industrialization index, GDP per capita, democratic index)? The results of these analyses can be found in the Supplement (Section 2).

## Pre-registration deviations

In our pre-registration document (registered on 16th September, 2024), we also planned to explore whether any of the effects were moderated by video length. Furthermore, we collected information on whether participants believed they had seen, not seen, or were uncertain whether they had seen the video in question before. While we were unable to include these variables as moderators because participants in the control group could not provide comparable information, we instead analyzed whether there were any significant differences in our main outcome variables by video length and whether participants believed they had seen the videos before. The results for these analyses can be found in the Supplement (Sections 5 and 6).

**Table 1 | Sample Quota Information for all 13 Surveys**

| Survey | Gender | | Age | | Total | Power (*d*) | |
|---|---|---|---|---|---|---|---|
| | Male | Female | 45–55 | 56+ | | 0.90 | 0.80 |
| Belgium (Flemish) | 765 | 706 | 696 | 777 | 1473 | 0.20 | 0.17 |
| Belgium (French) | 355 | 368 | 349 | 376 | 725 | 0.28 | 0.25 |
| Bulgaria | 264 | 287 | 352 | 199 | 551 | 0.32 | 0.28 |
| France[a] | 1438 | 1462 | 1395 | 1507 | 2903 | 0.16 | 0.14 |
| Germany[a] | 1473 | 1436 | 1227 | 1692 | 2919 | 0.16 | 0.14 |
| Hungary | 715 | 722 | 730 | 716 | 1446 | 0.20 | 0.17 |
| Ireland | 270 | 274 | 273 | 273 | 546 | 0.32 | 0.28 |
| Italy[a] | 1427 | 1441 | 1489 | 1389 | 2878 | 0.16 | 0.14 |
| Lithuania | 277 | 270 | 268 | 279 | 547 | 0.32 | 0.28 |
| Netherlands | 726 | 746 | 715 | 762 | 1477 | 0.20 | 0.17 |
| Poland | 712 | 731 | 860 | 586 | 1447 | 0.20 | 0.17 |
| Romania | 718 | 715 | 797 | 645 | 1442 | 0.20 | 0.17 |
| Spain | 706 | 709 | 723 | 698 | 1421 | 0.20 | 0.18 |
| Total | 9846 | 9867 | 9874 | 9899 | 19,775 | 0.22 | 0.20 |

*Note.* [a]Indicates that seven conditions were included in the survey; all other surveys included four conditions. Values from "Gender" to "Total" are sample size *N*s for each survey, values under "Power" are the minimum Cohen's *d*s that each sample is powered for based on Sensitivity power analyses with powers of 0.90 and 0.80, respectively. Total Cohen's *d* corresponds to average Cohen's *d* in the two respective power analyses.

## Methods
### Participants

Roozenbeek et al.[13] found in their lab studies that the average effect of prebunking videos on improving technique discernment (termed "technique recognition" by the authors) of relevant content was strong *d* = 0.50. A priori power analysis using the pwr package in R indicated that to detect this effect size with a power of 0.90 correcting for multiple comparisons with four conditions, a minimum sample size of *N* = 432 is required (*N* = 108 per condition). The same power analysis for an experiment with seven conditions indicated that a minimum sample size *N* = 860 is required (*N* = 123 per condition). While smaller effect sizes were obtained for manipulation discernment (termed "trustworthiness") by Roozenbeek et al.[13], we chose to power for this larger effect due to financial and resource constraints, as well as our plan to run aggregate analyses on all surveys. Nevertheless, we aimed to power sample sizes for effect sizes that were as small as possible. Therefore, only studies with a minimum sample size of *N* = 2800 included seven conditions (minimum *N* = 400 per condition). All other studies included four conditions (*N*s = 136 to 369 per condition).

All 13 surveys were distributed through *Bilendi's* data collection system from 17th September 2024 until 14th October 2024 (roughly three months after the June 2024 EU elections), returning 34,429 responses. Quotas were set so that in each survey, roughly half of participants would be male and the other half female, as well as quotas specifying half of the sample to be aged 45–55 and the other half aged 56+ ($M_{age}$ = 56.98, $SD_{age}$ = 8.56). After data exclusions for quota fills (*N* = 9170) and attention check failures (*N* = 7792; *M* = 29.56%, *SD* = 4.47%; *Min.* = 21.07%, *Max.* = 38.56%), a total sample size of 19,773 participants was obtained (see Table 1 for sample quota information).

When asked "Do you remember watching this video online in the past three months, or a video from the same campaign?", *N* = 112 had seen the scapegoating video before, *N* = 2718 had not seen the scapegoating video before, and *N* = 136 were unsure whether they had seen the scapegoating video before. For the decontextualization video, *N* = 168 had seen the video before, *N* = 4857 had not seen the video before, and *N* = 238 were unsure whether they had seen the video before. For the discrediting video, *N* = 126

had seen the video before, $N = 4071$ had not seen the video before, and $N = 209$ were unsure whether they had seen the video before. Although we could not compare the inoculation effects between participants who had seen, had not seen, or were unsure of whether they had seen the videos before because awareness of having seen the control video before was irrelevant, we compared the means of each outcome variable between these participants (see Supplement, Section 5). In general, participants who had not seen the videos before appeared to show better detection of manipulation and non-manipulation, were less likely to report willingness to share content overall, and were less confident in detecting manipulation compared to those who had seen or were unsure whether they had seen the videos before.

## Design and procedure
All surveys included the same experimental design, except for the number of experimental conditions participants could be allocated to. After being informed and providing their consent, they reported their socio-demographic information and completed the first attention check. Participants who failed this attention check were immediately excluded from the survey. Next, participants completed the measure of general manipulation discernment and were then randomly allocated to one of the possible conditions.

The conditions included one control condition or three possible experimental conditions. In the control condition, participants watched a 60s video of Lionel Messi's football career highlights. In the experimental conditions, participants watched a 50s video prebunking one of the following manipulation tactics: scapegoating, decontextualization, or discrediting (see OSF repository: https://osf.io/tkymv/overview?view_only=7486ede8ede1418f9a95855e7da5e5ac for videos). In the surveys distributed to residents of France, Germany, and Italy, three additional experimental conditions were included (totaling seven conditions in each survey), which presented the shorter 20s prebunking videos of the three respective manipulation tactics. After watching the video, participants were asked whether they had seen the video before and how likely they would be to share the video in their social networks, before answering the second attention check. Participants who failed this attention check were immediately excluded from the survey.

Participants were then presented with the item rating task. In the item rating task, there were 48 possible items that could be presented to participants. 24 contained manipulative content (eight used scapegoating, eight used decontextualization, and eight used discrediting), and the other 24 acted as non-manipulative counterpart items for each of the manipulative items (see OSF repository: https://osf.io/tkymv/overview?view_only=7486ede8ede1418f9a95855e7da5e5ac for item stimuli). Each non-manipulative counterpart item included information on the same topic as its manipulative counterpart item (e.g., online shopping addiction). In the control condition, participants were presented with a total of 18 items in the item rating task (six items for each of the three respective tactics).

The items were grouped by tactic and presented in randomized order. For each counterpart, participants were randomly presented with either the manipulative or the non-manipulative counterpart. In the experimental conditions, participants were presented with a total of 16 items. The first eight pertained to the manipulation tactic prebunked in the video they viewed, and then they were presented with four items from each of the other two respective tactics, grouped by tactic in randomized order. The number of manipulative and non-manipulative posts within each category was also completely randomized. For each item, participants rated their perceived manipulativeness of the content (manipulativeness assessments), their willingness to share the content within their social network (willingness to share), and a multiple-choice list of four options (technique recognition). The measure of technique recognition always included the correct manipulation tactic (e.g., "Scapegoating"), two decoy answers, and "None of the above" to indicate no manipulation. The order of the respective manipulation tactic answer and two decoy answers were always randomized, while the "None of the above" option was always presented at the bottom.

Next, participants indicated their confidence in spotting manipulative content before moving onto the final section of the study. In this final section, participants were presented with the measures of voting behavior in the June 2024 EU elections and digital literacy in random order. Participants were then presented with the qualitative measures of their feedback and reported behavior change as a result of watching the videos, alongside the multiple-choice questions of who they would trust to produce the content and what else they would like to learn in random order (for the purposes of another project). Finally, participants completed the political tolerance measure before being fully debriefed and financially compensated.

## Materials
In each of the longer inoculation videos, participants viewed a 50s animated video in their native language with embedded subtitles (see OSF repository for full stimuli: https://osf.io/tkymv/overview?view_only=7486ede8ede1418f9a95855e7da5e5ac). Each video followed the standard inoculation framework: 1) an affective forewarning of the threat of manipulation, 2) a pre-emptive refutation of the specific manipulation tactic (including example *microdoses* of what the manipulation tactic might look like), and 3) a summary of the message and call to action. In each of the shorter 20s videos, this framework was compressed so that fewer examples were presented, but the same information was communicated.

## Measures
Because the intervention directly targeted participants' ability to detect manipulation, pooling post-test data across experimental conditions could distort internal consistency estimates by combining groups with systematically different response patterns. To avoid this and to provide reliability estimates that reflect the unaltered measurement properties of the scales, Cronbach's alpha values for the main outcome variables—manipulativeness assessments, technique recognition, willingness to share, and confidence in detecting manipulation—are reported based on the control group alone.

**Covariates**. Participants indicated their gender ("Male", "Female", "Non-binary", "Other", or "Prefer not to say"), highest level of educational attainment ($1 = No\ formal\ education\ above\ age\ 16$, to $6 = Doctorate$; $M = 3.55$, $SD = 1.22$), age, and political ideology ($1 = Very\ left\text{-}wing$, to $7 = Very\ right\text{-}wing$; $M = 4.12$, $SD = 1.38$). The ethnicity of participants was not recorded.

**General manipulation discernment ability**. General manipulation discernment ability ($M = 7.37$, $SD = 2.01$; $\alpha = 0.35$) was measured using Maertens et al.'s[42] Manipulative Online Content Recognition Inventory (MOCRI), which includes six generally manipulative items (e.g., "You don't really care about lowering crime in the city, you just want people to vote for you") and six generally non-manipulative items (e.g., "Airline's 'Christmas Miracle' video goes viral on social media"). Participants were asked to use a binary scale to determine whether each item was "Manipulative" or "Non-manipulative". Correct detection of both the manipulative and non-manipulative items was summed to create a measure of general manipulation discernment ability. While the internal reliability of this measure was considerably low, the analysis indicated no particularly problematic items for removal. Furthermore, Maertens et al.[42] confirmed its reliability through many other psychometric analyses. Therefore, we decided to still include this measure as an exploratory moderator while taking note of its low internal consistency.

**Manipulativeness assessments**. Manipulativeness assessments were collected by presenting participants with the statement "This [statement/headline] is manipulative" under each piece of content in the item rating task, with a response scale from 1 (*Strongly disagree*) to 7 (*Strongly agree*). Manipulativeness assessments of the manipulative scapegoating (e.g., "Local businesses failing because of teens' online shopping addiction!"; $M = 4.49$, $SD = 1.64$; $\alpha = 0.90$), decontextualization (e.g., "You see a

peaceful photo with the headline "Violent protest against the law!""; $M = 4.48$, $SD = 1.63$; $\alpha = 0.78$), and discrediting ("Don't trust that economist about inflation - they were fined for reckless driving"; $M = 4.69$, $SD = 1.86$; $\alpha = 0.91$) content was calculated by averaging the manipulativeness assessments of the manipulative items for each of the three respective manipulation tactics. Manipulativeness assessments of the non-manipulative scapegoating (e.g., "Around 5.8% of the US general population currently suffers from shopping addiction."; $M = 3.79$, $SD = 1.56$; $\alpha = 0.70$), decontextualization (e.g., "You see a photo of protestors with the headline "Protests over new laws""; $M = 3.80$, $SD = 1.57$; $\alpha = 0.61$), and discrediting ("Economists explain how recessions occur"; $M = 3.59$, $SD = 1.59$; $\alpha = 0.58$) content was calculated by averaging the manipulativeness assessments of the non-manipulative items for each of the three respective manipulation tactics. Manipulation discernment of the scapegoating ($M = 0.63$, $SD = 2.02$; $\alpha = 0.80$), decontextualization ($M = 0.65$, $SD = 1.91$; $\alpha = 0.87$), and discrediting ($M = 1.05$, $SD = 2.45$; $\alpha = 0.80$) content was calculated by subtracting the manipulativeness assessments of the non-manipulative items from the manipulativeness assessments of the manipulative items for each of the three respective manipulation tactics, so that higher scores indicated stronger manipulation discernment.

**Technique recognition.** Technique recognition of manipulative and non-manipulative content was measured by presenting participants with the statement "This [statement/headline] uses…" and four multiple-choice response options. For each item set, there was a relevant manipulation tactic (e.g., "Scapegoating", for the scapegoating block, even when non-manipulative content had been presented) and "None of the above", with two decoy answers (e.g., "Appeal to conflict"). Technique recognition of the manipulative scapegoating ($M = 52.87\%$, $SD = 37.96\%$; $\alpha = 0.70$), decontextualization ($M = 36.48\%$, $SD = 36.02\%$; $\alpha = 0.60$), and discrediting ($M = 58.11\%$, $SD = 38.10\%$; $\alpha = 0.73$) content was measured by calculating the proportion of correct responses for each of the three respective manipulation tactics. Technique recognition of the non-manipulative scapegoating ($M = 44.45\%$, $SD = 38.54\%$; $\alpha = 0.77$), decontextualization ($M = 38.84\%$, $SD = 37.26\%$; $\alpha = 0.75$), and discrediting ($M = 57.85\%$, $SD = 78.29\%$; $\alpha = 0.75$) content was measured by calculating the proportion of times "None of the above" was correctly selected for the non-manipulative items for each of the three respective manipulation tactics. Technique discernment for the scapegoating ($M = 8.49\%$, $SD = 55.41\%$; $\alpha = 0.70$), decontextualization ($M = -2.42\%$, $SD = 54.93\%$; $\alpha = 0.60$), and discrediting ($M = 0.29\%$, $SD = 87.90\%$; $\alpha = 0.73$) content was measured by subtracting the proportion of correct responses for the non-manipulative content from the proportion of correct responses for the manipulative content, so that higher scores indicated greater technique discernment.

**Willingness to share.** Willingness to share was measured by presenting participants with the statement "I would share this [statement/headline] with people in my network" under each piece of content in the item rating task, with a response scale from 1 (*Strongly disagree*) to 7 (*Strongly agree*). Willingness to share the manipulative scapegoating ($M = 2.76$, $SD = 1.79$; $\alpha = 0.93$), decontextualization ($M = 2.71$, $SD = 1.71$; $\alpha = 0.93$), and discrediting ($M = 2.47$, $SD = 1.72$; $\alpha = 0.96$) content was calculated by averaging willingness to share the manipulative items for each of the three respective manipulation tactics. Willingness to share the non-manipulative scapegoating ($M = 3.00$, $SD = 1.80$; $\alpha = 0.93$), decontextualization ($M = 2.96$, $SD = 1.76$; $\alpha = 0.88$), and discrediting ($M = 2.91$, $SD = 1.76$; $\alpha = 0.85$) content was calculated by averaging willingness to share the non-manipulative items for each of the three respective manipulation tactics. Sharing decisions for the scapegoating ($M = 0.23$, $SD = 1.31$; $\alpha = 0.96$), decontextualization ($M = 0.24$, $SD = 1.20$; $\alpha = 0.96$), and discrediting ($M = 0.38$, $SD = 1.29$; $\alpha = 0.97$) content was calculated by subtracting willingness to share the manipulative items from willingness to share the non-manipulative items for each of the three respective

manipulation tactics, so that higher scores indicated better sharing decisions.

**Confidence in detecting manipulation.** Confidence in detecting manipulation ($M = 4.57$, $SD = 1.22$) was measured by presenting participants with the statement "I am confident in my ability to detect manipulative content [such as scapegoating/decontextualization/discrediting]". Participants used a response scale from 1 (*Strongly disagree*) to 7 (*Strongly agree*).

**Voting behavior in the June 2024 EU elections.** Voting behavior in the June 2024 EU elections was measured by asking participants, "Did you vote in June's EU elections?" with the response options *Yes* ($N = 15,800$) or *No* ($N = 3371$).

**Digital literacy.** Digital literacy ($M = 3.94$, $SD = 0.87$; $\alpha = 0.87$) was measured by presenting participants with eight items, five of which were taken from Van Deurson et al.'s[43] operational skills subscale from the digital literacy scale (e.g., "I know how to open downloaded files"), and three were original items developed for the purposes of this research and to improve the representation of digital literacy in a more modern context since 2014 when the original digital literacy scale was published (e.g., "I know how to create and collaborate on digital documents"). Participants were asked to indicate how well each of the items represented themselves using a response scale from 1 (*Not at all true of me*) to 5 (*Very true of me*).

**Political tolerance.** Political tolerance ($M = 4.97$, $SD = 0.97$; $\alpha = 0.63$) was measured with Dunwoody and Funke's[35] six-item political tolerance scale (e.g., "People who disagree with me politically deserve the same rights as I do") with a response scale from 1 (*Strongly disagree*) to 7 (*Strongly agree*).

**WEIRDness of the sample nations.** Western vs. non-Western was coded by reporting the longitude of the center point of each nation that took part in the surveys. Positive scores indicated more Eastern European nations, and negative scores indicated more Western nations ($M = 10.11$, $SD = 8.87$; $Min. = -8.24$, $Max. = 25.49$). Education index ($M = 0.89$, $SD = 0.05$; $Min. = 0.77$, $Max. = 0.94$) was coded using values from the United Nations Development Programme[44]. Industrialization index ($M = 0.78$, $SD = 6.41$; $Min. = -4.6$, $Max. = 34.9$) was coded using Eurostat's[45] industrial production statistics. GDP per capita ($M = 37,550.04$, $SD = 20,490.20$; $Min. = 5540$, $Max. = 106,060$) was coded using values from the International Monetary Fund's[46] report of GDP per capita. Democratic index ($M = 7.85$, $SD = 0.79$; $Min. = 6.49$, $Max. = 9.13$) was coded using values from the Economist Intelligence Unit's[47] Democracy Index.

## Ethics
Full ethical approval for this project was granted by the King's College London Research Ethics Committee, reference number: MRA-23/24-45147.

## Ethics and inclusion statement
This multi-country study was conducted across twelve EU nations through collaborations between researchers based in the UK, the Netherlands, and the United States, with support from Moonshot and Google Jigsaw. While some nations included in the surveys did not have resident co-authors, local expertise was incorporated at multiple stages of the project. Specifically, residents of each nation contributed to the development and testing of the prebunking video content on YouTube, and native speakers from each country translated, adapted, and piloted the survey materials to ensure linguistic and cultural relevance. All materials were designed to address misinformation relevant to each country's 2024 EU election context, ensuring local applicability. No part of the research posed risks of stigmatization, discrimination, or personal harm to participants. Data collection

complied with the General Data Protection Regulation (GDPR), and all participants provided informed consent before taking part.

## Analysis plan

To evaluate the effectiveness of our inoculation videos, we conducted three-level multilevel models ($p$-values two-tailed by default), nesting surveys within survey language and then within the country of the sample. The condition variable (seven levels: three long inoculation videos, three short inoculation videos, one control) was entered as the predictor, with the respective dependent variables as outcomes. Our main analyses assessed the effects of the inoculation videos (vs. control) on manipulation discernment (higher manipulativeness ratings for relevant manipulative vs. non-manipulative content), technique discernment (correct identification of relevant manipulative vs. non-manipulative content), sharing decisions (greater willingness to share relevant non-manipulative vs. manipulative content), and confidence in detecting relevant manipulation for each of the three tactics. Regression coefficients are reported to represent the estimated effects of predictors while controlling for the model intercept(s) and random effects, and Cohen's $d$ was calculated from estimated marginal means using the eff_size() function in emmeans, which standardizes pairwise mean differences by the residual standard deviation from the fitted mixed-effects model. To evaluate the strength of evidence for null vs. alternative hypotheses in cases of non-significant findings, we conducted Bayesian independent-samples $t$-tests using the default JZS prior (Cauchy scale $r = 0.71$) as implemented in the BayesFactor package in R. These analyses allowed us to quantify the degree of support for the null hypothesis, supplementing traditional $p$-values with continuous measures of evidence strength. The relevant contrasts were specified for each relevant experimental condition vs. the control group. As the Bayes Factors in the *Bayes-Factor* package are computed via analytic solutions rather than simulation, no Markov chain Monte Carlo settings were applied.

For detailed results on unrelated content, see the Supplement (Section 1). Summaries of the results for significant interaction effects between the inoculation videos (vs. control) and moderators for relevant content are reported in their respective sections here. For full statistical details, including results for interactions on different outcomes, refer to the Supplement (Section 2). For violin plots of the means for each outcome variable between conditions, please refer to the Supplement (Section 7). For brevity, only the significant interaction effects for manipulation discernment, technique discernment, sharing decisions, and confidence in detecting manipulation are reported here. For details of significant interaction effects on other outcome variables, please refer to the Supplement (Section 2). To ensure model convergence, interaction effects were tested using single-level models with the survey as the only random factor. All data and inoculation videos are available in the OSF repository: https://osf.io/tkymv/overview?view_only=7486ede8ede1418f9a95855e7da5e5ac, and the experimental designs, sample size plan, and hypotheses are fully pre-registered at: https://osf.io/kjbdv/overview?view_only=901e19c4d8bf4ae9928f1a5dc688b554.

Given the large sample size and use of multilevel models, formal tests of normality and homogeneity of variance were not conducted. Mixed-effects models are robust to moderate deviations from normality, and large samples ensure asymptotic validity of parameter estimates and standard errors. Data distribution was assumed to be normal, but this was not formally tested.

## Results

### Manipulativeness assessments

**Scapegoating**. As a robustness check, we conducted a paired samples $t$-test comparing manipulativeness assessments of the manipulative vs. non-manipulative scapegoating content only on participants in the control group. The manipulative scapegoating content was perceived as significantly more manipulative, $M = 4.49$, $SD = 1.71$, than the non-manipulative scapegoating content, $M = 3.86$, $SD = 1.50$, $t(3,777) = 18.84$, $p < .001$, $d_z = 0.31$, 95% CI [0.28, 0.35], validating this as a measure of manipulation discernment.

Manipulation discernment. The positive effect of the long scapegoating video (vs. control) on manipulation discernment of the scapegoating content was significant, $t(17,900) = 3.49$, $p < .001$, $d = 0.08$, 95% CI [0.03, 0.12], but this was not the case for the short scapegoating video, $t(16,560) = 0.77$, $p = .439$, $d = 0.03$, 95% CI [−0.04, 0.10] (vs. control; the Bayes Factor for this effect indicated very weak evidence for the null hypothesis over the alternative, $BF_{01} = 1.34$, $BF_{10} = 0.75$; see Fig. 1). The positive effect of the long scapegoating video (vs. control) on manipulation discernment of the scapegoating content was significantly moderated by both political tolerance, $b = 0.11$, $SE = 0.05$, $t(17,270) = 2.17$, $p = .030$, and intentions to share the video within one's social network, $b = 0.09$, $SE = 0.03$, $t(17,200) = 3.29$, $p < .001$, such that the effect was only positive and significant at higher and more moderate levels of both political tolerance and intentions to share the video (see Supplement, Section 2, Fig. S8).

Manipulative content. The long scapegoating video (vs. control) significantly increased manipulativeness assessments of the manipulative scapegoating content, $t(18,640) = 5.08$, $p < .001$, $d = 0.11$, 95% CI [0.07, 0.16], but the effect of the short scapegoating video (vs. control) was non-significant, $t(12,120) = 0.78$, $p = .434$, $d = 0.03$, 95% CI [−0.04, 0.10] (the Bayes Factor for this effect also indicated strong evidence for the null hypothesis over the alternative, $BF_{01} = 24.1$, $BF_{10} = 0.04$; see Fig. 1). The long scapegoating video (vs. control) also significantly increased manipulativeness assessments of the manipulative decontextualization and discrediting content (see Supplement, Section 1).

Non-manipulative content. The long, $t(18,640) = 0.89$, $p = .374$, $d = 0.02$, 95% CI [−0.02, 0.06], and short scapegoating videos (vs. control), $t(18,010) = 0.02$, $p = .986$, $d = 0.01$, 95% CI [−0.07, 0.07], did not significantly alter manipulativeness assessments of the non-manipulative scapegoating content (the Bayes Factor for the effect of the long scapegoating video (vs. control) indicated very strong evidence for the null hypothesis over the alternative, $BF_{01} = 26.4$, $BF_{10} = 0.04$, and the Bayes Factor for the effect of the short scapegoating video (vs. control) indicated only anecdotal evidence for the null hypothesis over the alternative, $BF_{01} = 0.60$, $BF_{10} = 1.67$; see Fig. 1).

**Decontextualization**. As a robustness check, we conducted a paired samples $t$-test comparing manipulativeness assessments of the manipulative vs. non-manipulative decontextualization content only on participants in the control group. The manipulative decontextualization content was perceived as significantly more manipulative, $M = 4.46$, $SD = 1.65$, than the non-manipulative decontextualization content, $M = 3.83$, $SD = 1.52$, $t(3779) = 20.01$, $p < .001$, $d_z = 0.34$, 95% CI [0.31, 0.38], validating this as a measure of manipulation discernment.

Manipulation discernment. The positive effect of the long decontextualization video (vs. control) on manipulation discernment of the decontextualization content was significant, $t(17,990) = 4.47$, $p < .001$, $d = 0.10$, 95% CI [0.06, 0.15], but this was not the case for the short decontextualization video (vs. control), $t(13,330) = 0.90$, $p = .368$, $d = 0.03$, 95% CI [−0.04, 0.10] (the Bayes Factor for this effect indicated moderate evidence for the null hypothesis over the alternative, $BF_{01} = 3.65$, $BF_{10} = 0.27$; see Fig. 1). The long decontextualization video (vs. control) also significantly increased manipulation discernment of the scapegoating and discrediting content (see Supplement, Section 1). Furthermore, the short decontextualization video (vs. control) also significantly increased manipulation discernment of the scapegoating content (see Supplement, Section 1). The positive effect of the long decontextualization video (vs. control) on manipulation discernment of the decontextualization content was significantly moderated by intentions to share the video within one's social network, $b = 0.09$, $SE = 0.02$, $t(17,990) = 4.03$, $p < .001$, such that the effect on manipulation discernment was positive and significant when intentions were higher or more moderate, but not when intentions were lower (see Supplement, Section 2, Fig. S30).

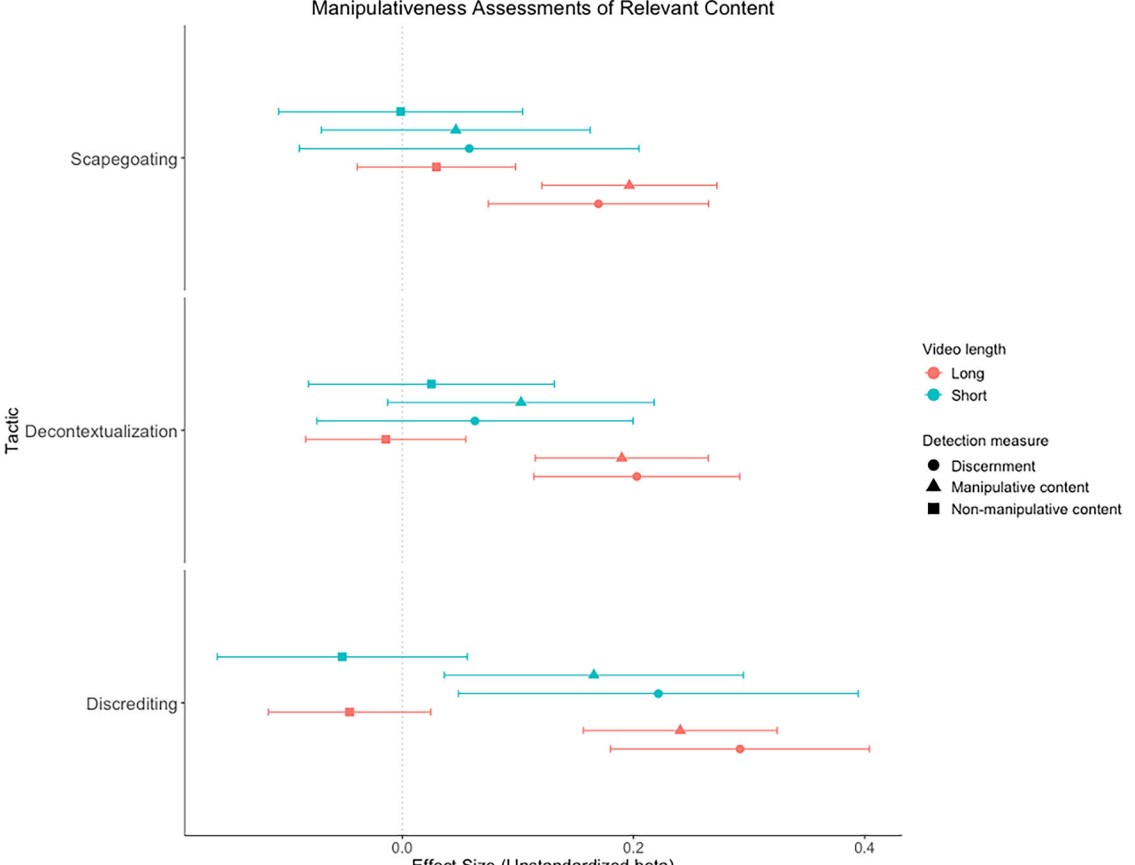

Manipulativeness Assessments of Relevant Content

**Fig. 1 | Forest plot of unstandardized beta coefficients and 95% confidence intervals for the multilevel model effects of all videos on manipulation discernment and manipulativeness assessments of the relevant manipulative and non-manipulative content for the three respective manipulation tactics.** This Shows that the Long Videos Significantly Increased Relevant Manipulation Discernment and Manipulativeness Assessments of the Relevant Manipulative Content Compared to the Control Group for all Tactics, and Similar Effects were Indicated for the Short Discrediting Video. None of the Videos Significantly Altered Manipulativeness Assessments of the Relevant Non-Manipulative Content Compared to the Control Group. *Note. N* = 17,898 to 18,738. Positive effect sizes denote higher manipulativeness assessments and discernment in the experimental conditions compared to the control group.

Manipulative content. The long decontextualization video (vs. control) significantly increased manipulativeness assessments of the manipulative decontextualization content, $t(18,720) = 4.97$, $p = < .001$, $d = 0.11$, 95% CI [0.07, 0.16], but the effect of the short decontextualization video (vs. control) was non-significant, $t(16,130) = 1.75$, $p = .081$, $d = 0.06$, 95% CI [−0.01, 0.13] (the Bayes Factor for this effect indicated very strong evidence for the null hypothesis over the alternative, $BF_{01} = 24.0$, $BF_{10} = 0.04$; see Fig. 1). The long decontextualization video (vs. control) also significantly increased manipulativeness assessments of the manipulative scapegoating and discrediting content (see Supplement, Section 1).

Non-manipiulaitve content. The long, $t(18,720) = 0.39$, $p = .694$, $d = −0.01$, 95% CI [−0.05, 0.04], and short decontextualization videos (vs. control), $t(16,300) = 0.56$, $p = .579$, $d = 0.02$, 95% CI [−0.05, 0.09], did not significantly alter manipulativeness assessments of the non-manipulative decontextualization content (the Bayes Factor for the effect of the long decontextualization video (vs. control) indicated very strong evidence for the null hypothesis over the alternative, $BF_{01} = 36.5$, $BF_{10} = 0.03$, and the Bayes Factor for the effect of the short decontextualization video (vs. control) indicated moderate evidence for the null hypothesis over the alternative, $BF_{01} = 3.1$, $BF_{10} = 0.32$; see Fig. 1). The long decontextualization video (vs. control) also significantly increased manipulativeness assessments of the non-manipulative scapegoating content (see Supplement, Section 1).

**Discrediting**. As a robustness check, we conducted a paired samples *t*-test comparing manipulativeness assessments of the manipulative vs.

non-manipulative discrediting content only on participants in the control group. The manipulative discrediting content was perceived as significantly more manipulative, $M = 4.66$, $SD = 1.89$, than the non-manipulative discrediting content, $M = 3.63$, $SD = 1.53$, $t(3,759) = 25.52$, $p < .001$, $d_z = 0.43$, 95% CI [0.40, 0.46], validating this as a measure of manipulation discernment.

Manipulation discernment. The positive effects of the long, $t(17,880) = 5.11$, $p < .001$, $d = 0.12$, 95% CI [0.07, 0.16], and short discrediting videos (vs. control), $t(15,830) = 2.51$, $p = .012$, $d = 0.09$, 95% CI [0.02, 0.16], on manipulation discernment of the discrediting content were significant (see Fig. 1). The positive effect of the long discrediting video (vs. control) on manipulation discernment of the discrediting content was also significantly moderated by the GDP per capita of nations, $b = −0.01$, SE = 0.01, $t(17,490) = −2.15$, $p = .032$, such that the effect was only positive and significant at lower and more moderate levels of GDP per capita, but not at higher levels (see Supplement, Section 2, Fig. S54). There was a significant interaction between the short discrediting video (vs. control) and age when predicting manipulation discernment of the discrediting content, $b = −0.021$, $SE = 0.010$, $t(17,490) = −2.05$, $p = .040$, such that the effect was only positive and significant among younger participants, but not among participants whose age was closer to the mean or older participants (see Supplement, Section 2, Fig. S55). The positive effects of both the long, $b = 0.15$, SE = 0.03, $t(17,880) = 5.28$, $p < .001$, and short discrediting videos (vs. control), $b = 0.15$, SE = 0.04, $t(17,750) = 3.62$, $p < .001$, on manipulation discernment of the discrediting content were also moderated by intentions

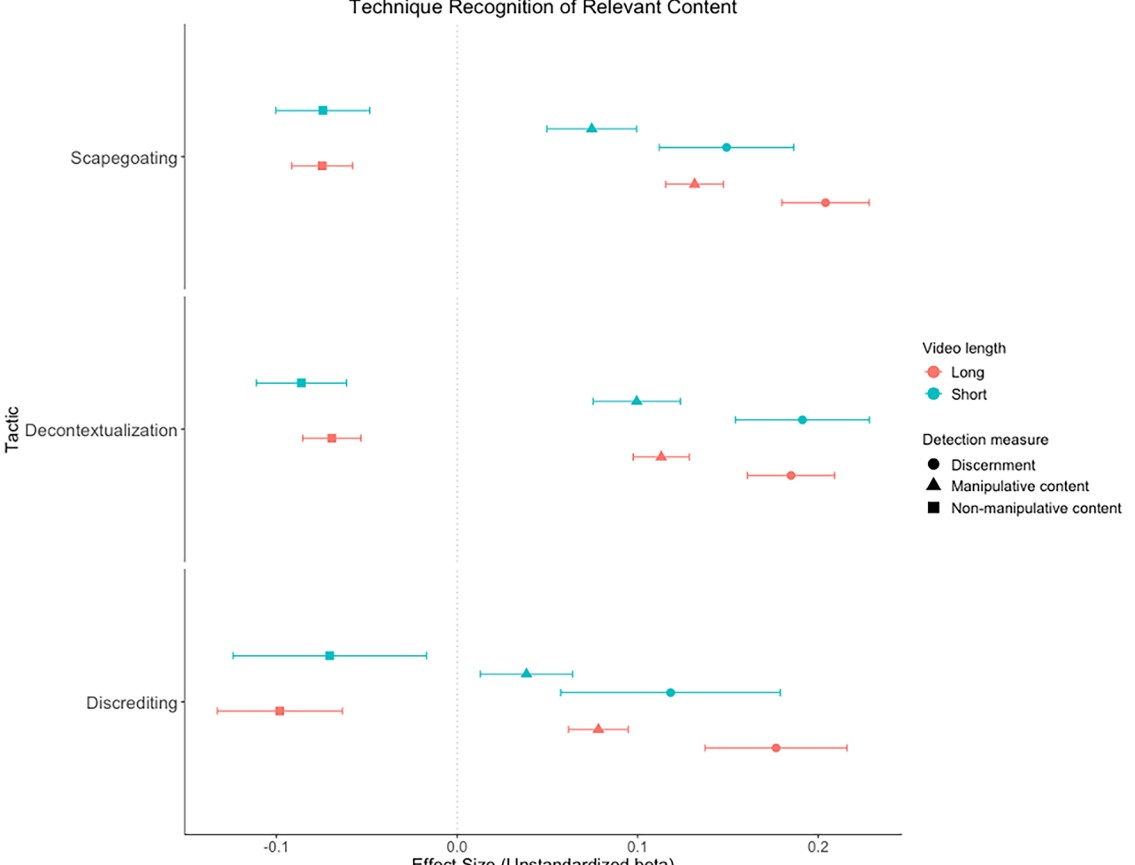

**Fig. 2 | Forest plot of unstandardized beta coefficients and 95% confidence intervals for the multilevel model effects of all videos on technique discernment and technique recognition of the relevant manipulative and non-manipulative content for the three respective manipulation tactics.** This Shows that the Long Videos Significantly Increased Technique Discernment and Detection of the Relevant Manipulative Content, and Reduced Technique Recognition of the Relevant Non-Manipulative Content Compared to the Control Group for all Tactics. The Short Videos also Significantly Increased Technique Recognition of the Relevant Manipulative Content and Reduced Technique Recognition of the Relevant Non-Manipulative Content Compared to the Control Group. The Effects of the Short Videos on Manipulation Discernment of the Relevant Content Compared to the Control Group were Non-Significant. *Note. N* = 17,917 to 18,738. Positive effect sizes denote higher correct detection of manipulative or non-manipulative content in the experimental conditions compared to the control group. Negative effect sizes denote lower correct detection of manipulative or non-manipulative content in the experimental conditions compared to the control group.

to share the video, such that the effects of the long and short discrediting videos (vs. control) were only positive and significant when intentions to share were higher or more moderate, but not when intentions were lower (see Supplement, Section 2, Fig. S56).

Manipulative content. The long, $t(18,660) = 5.62, p < .001, d = 0.13, 95\%$ CI [0.08, 0.17], and short discrediting videos (vs. control), $t(15,830) = 2.51, p = .012, d = 0.09, 95\%$ CI [0.02, 0.16], significantly increased manipulativeness assessments of the manipulative discrediting content (see Fig. 1). The long discrediting video (vs. control) also significantly increased manipulativeness assessments of the manipulative scapegoating content and manipulation discernment of the scapegoating content (see Supplement, Section 1).

Non-manipulative content. The long, $t(18,640) = 1.25, p = .212, d = −0.03, 95\%$ CI [−0.07, 0.02], and short discrediting videos (vs. control), $t(15,930) = 0.89, p = .374, d = −0.03, 95\%$ CI [−0.10, 0.04], did not significantly alter manipulativeness assessments of the non-manipulative discrediting content (the Bayes Factor for the effect of the long discrediting video (vs. control) indicated strong evidence for the null hypothesis over the alternative, $BF_{01} = 16.0, BF_{10} = 0.06$, and the Bayes Factor for the effect of the short discrediting video (vs. control) indicated very weak evidence for the alternative hypothesis over the null, $BF_{01} = 0.84, BF_{10} = 1.20$; see Fig. 1). The long and short discrediting videos (vs. control) also significantly decreased

manipulativeness assessments of the non-manipulative scapegoating content (see Supplement, Section 1).

### Technique recognition
**Scapegoating.** The positive effects of the long, $t(17,600) = 16.52, p < .001, d = 0.38, 95\%$ CI [0.34, 0.43], and short scapegoating videos (vs. control), $t(17,570) = 7.84, p < .001, d = 0.28, 95\%$ CI [0.21, 0.34], on technique discernment of the scapegoating content were small-to-medium and significant (see Fig. 2). The long and short scapegoating videos (vs. control) also significantly increased technique discernment of the decontextualization and discrediting content (see Supplement, Section 1). The positive effect of the long scapegoating video (vs. control) on technique discernment of the scapegoating content was also significantly moderated by Western (vs. non-Western) European nations, $b = −0.01, SE = 0.01, t(19,370) = 3.53, p < .001$, the education indices of nations, $b = 0.30, SE = 0.10, t(19,370) = 2.91, p = .004$, the GDP per capita of nations, $b = 0.01, SE = 0.01, t(19,370) = 2.52, p = .012$, and the democratic indices of nations, $b = 0.03, SE = 0.01, t(19,370) = 4.13, p < .001$, such that the effect was only positive and significant among more Western or central European nations, and nations with more moderate or higher education indices, GDP per capita, or democratic indices (see Supplement, Section 2, Figs. S19–S22). The positive effect of the long scapegoating video (vs. control) on technique discernment of the scapegoating content was also significantly moderated by personal educational

attainment, $b = 0.01$, SE = 0.01, $t(19,370) = 2.26$, $p = .024$, such that the effect was only positive and significant among participants with more moderate or higher levels of educational attainment, but not for participants with lower educational attainment (see Supplement, Section 2, Fig. S23). The positive effect of the short scapegoating video (vs. control) on technique discernment of the scapegoating content was also moderated by intentions to share the video, $b = 0.02$, $SE = 0.01$, $t(17,900) = 2.11$, $p = .035$, such that the effect was strongest at at higher intentions, weaker at more moderate intentions, and weakest at lower levels of intentions to share (see Supplement, Section 2, Fig. S24).

Manipulative content. Both the long, $t(18,640) = 16.02$, $p < .001$, $d = 0.36$, 95% CI [0.32, 0.41], and short scapegoating videos (vs. control), $t(18,640) = 5.88$, $p < .001$, $d = 0.21$, 95% CI [0.14, 0.27], significantly increased the detection of manipulative scapegoating content (see Fig. 2). The long and short scapegoating videos (vs. control) also significantly increased technique recognition of the manipulative decontextualization and discrediting content (see Supplement, Section 1).

Non-manipulative content. However, both the long, $t(18,640) = 8.67$, $p < .001$, $d = -0.20$, 95% CI [−0.24, −0.15], and short scapegoating videos (vs. control), $t(17,920) = 5.60$, $p < .001$, $d = -0.19$, 95% CI [−0.26, −0.13], significantly decreased the detection of non-manipulative scapegoating content (see Fig. 2). The long and short scapegoating videos (vs. control) also significantly decreased technique recognition of the non-manipulative decontextualization and discrediting content (see Supplement, Section 1).

### Decontextualization
Technique discernment. The positive effects of the long, $t(17,690) = 15.05$, $p < .001$, $d = 0.35$, 95% CI [0.30, 0.39], and short decontextualization videos (vs. control), $t(17,630) = 10.09$, $p < .001$, $d = 0.36$, 95% CI [0.28, 0.42], on technique discernment of the decontextualization content were small-to-medium and significant (see Fig. 2). The long and short decontextualization videos (vs. control) also significantly increased technique discernment of the scapegoating and discrediting content (see Supplement, Section 1). The positive effect of the long decontextualization video (vs. control) on technique discernment of the decontextualization content was significantly moderated by the education indices of nations, $b = -0.20$, SE = 0.10, $t(19,440) = 2.09$, $p = .036$, such that it was only positive and significant at lower and more moderate levels of education indices (see Supplement, Section 2, Fig. S43). The positive effects of both the long, $b = 0.03$, SE = 0.00, $t(18,710) = 6.51$, $p < .001$, and short decontextualization videos (vs. control), $b = 0.02$, SE = 0.01, $t(18,710) = 3.98$, $p < .001$, on technique discernment of the decontextualization content were moderated by intentions to share the video within one's social network. For the long decontextualization video (vs. control), the effect was strongest at higher levels, weaker at more moderate levels, and weakest at lower levels of intentions to share (see Supplement, Section 2, Fig. S44). For the short decontextualization video (vs. control), the effect was only positive and significant at higher and more moderate levels of intentions to share, but not at lower levels (see Supplement, Section 2, Fig. S44).

Manipulative content. Both the long, $t(18,720) = 14.19$, $p < .001$, $d = 0.32$, 95% CI [0.28, 0.36], and short decontextualization videos (vs. control), $t(18,670) = 8.08$, $p < .001$, $d = 0.28$, 95% CI [0.21, 0.35], significantly increased the technique recognition of manipulative decontextualization content (see Fig. 2). The long and short decontextualization videos (vs. control) also significantly increased technique recognition of the manipulative scapegoating and discrediting content (see Supplement, Section 1).

Non-manipulative content. Both the long, $t(18,720) = 8.41$, $p < .001$, $d = -0.19$, 95% CI [−0.23, −0.15], and short decontextualization videos (vs. control), $t(18,580) = 6.76$, $p < .001$, $d = -0.24$, 95% CI [−0.30, −0.17], significantly decreased technique recognition of the non-manipulative decontextualization content (see Fig. 2). The long and short

decontextualization videos (vs. control) also significantly decreased technique recognition of the non-manipulative scapegoating and discrediting content (see Supplement, Section 1).

### Discrediting
Technique discernment. The positive effects of the long, $t(17,580) = 8.80$, $p < .001$, $d = 0.20$, 95% CI [0.16, 0.25], and short discrediting videos (vs. control), $t(16,470) = 3.82$, $p < .001$, $d = 0.14$, 95% CI [0.06, 0.20], on technique discernment of the discrediting content were small but significant (see Fig. 2). The long and short discrediting videos (vs. control) also significantly increased technique discernment of the scapegoating and decontextualization content (see Supplement, Section 1). The positive effect of the long discrediting video (vs. control) on technique discernment of the discrediting content was significantly moderated by both personal educational attainment, $b = 0.01$, SE = 0.01, $t(19,410) = 2.45$, $p = .014$, and political tolerance, $b = 0.02$, SE = 0.01, $t(19,060) = 2.54$, $p = .011$, such that the effect was only positive and significant at higher levels of these moderators (see Supplement, Section 2, Figs. S63-S64).

Manipulative content. Both the long, $t(18,660) = 9.28$, $p < .001$, $d = 0.21$, 95% CI [0.17, 0.25], and short discrediting videos (vs. control), $t(18,640) = 2.94$, $p = .004$, $d = 0.10$, 95% CI [0.03, 0.17], significantly increased the technique recognition of manipulative discrediting content (see Fig. 2). The long and short discrediting videos (vs. control) also significantly increased technique recognition of the manipulative scapegoating and decontextualization content (see Supplement, Section 1).

Non-manipulative content. Both the long, $t(17,880) = 5.55$, $p < .001$, $d = -0.13$, 95% CI [−0.17, −0.08], and short discrediting videos (vs. control), $t(16,090) = 2.58$, $p = .010$, $d = -0.09$, 95% CI [−0.16, −0.02], significantly decreased technique recognition of the non-manipulative discrediting content (see Fig. 2). The long and short discrediting videos (vs. control) also significantly decreased technique recognition of the non-manipulative scapegoating and decontextualization content (see Supplement, Section 1).

### Confidence in detecting manipulation
Scapegoating. The positive effect of the long scapegoating video (vs. control), $t(15,690) = 3.76$, $p < .001$, $d = 0.11$, 95% CI [0.06, 0.18], on confidence in detecting relevant manipulation was significant (see Fig. 3). In contrast, the effect of the short scapegoating video (vs. control) on confidence in detecting manipulation was non-significant, $t(15,690) = 1.61$, $p = .107$, $d = 0.07$, 95% CI [−0.01, 0.15] (the Bayes Factor for this effect indicated strong evidence for the null hypothesis over the alternative, $BF_{01} = 12.4$, $BF_{10} = 0.08$). The positive effect of the scapegoating video (vs. control) on confidence in detecting relevant manipulation was significantly moderated by the GDP per capita of nations, $b = -0.01$, SE = 0.01, $t(16,120) = 2.17$, $p = .030$, such that the effect was only positive and significant among nations with lower and more moderate GDP per capita, but not among nations with higher GDP per capita (see Supplement, Section 2, Fig. S77). The positive effect of the scapegoating video (vs. control) on confidence in detecting relevant manipulation was also significantly moderated by voting behavior, $b = 0.18$, SE = 0.09, $t(16,120) = 2.14$, $p = .033$, such that the effect was stronger among those who did not vote in the June 2024 EU elections and weaker among those who did vote (see Supplement, Section 2, Fig. S78).

Decontextualization. The negative effects of the long, $t(15,830) = 3.37$, $p < .001$, $d = -0.08$, 95% CI [−0.12, 0.03], and short decontextualization videos (vs. control), $t(15,520) = 2.01$, $p = .044$, $d = -0.07$, 95% CI [−0.15, 0.01], on confidence in detecting relevant manipulation were significant. Sensitivity power analysis indicated that the effect size magnitude for the effect of the short decontextualization video (vs. control) on confidence in detecting manipulation ($d = -0.07$) did not reach the minimum effect size for a power of 0.80 ($d = 0.08$). The effect of the short decontextualization video (vs. control) on confidence in detecting relevant

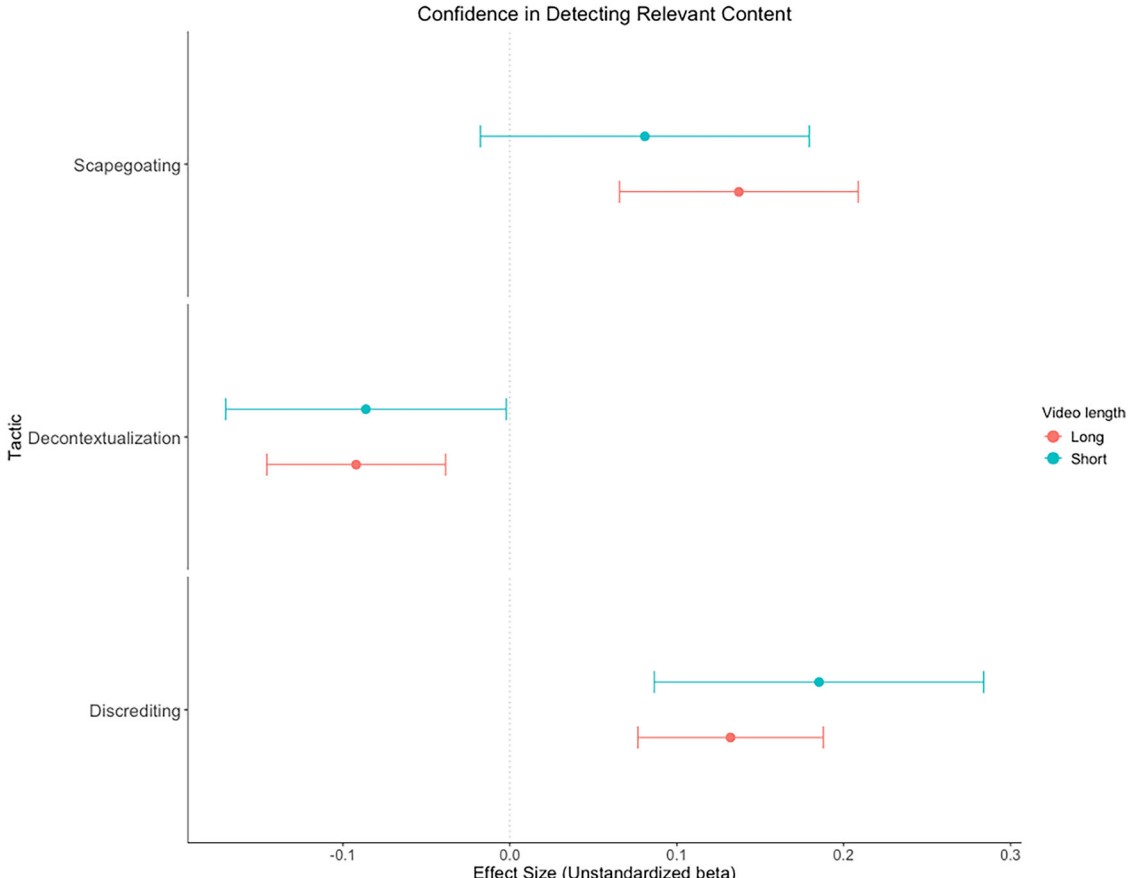

**Fig. 3 | Forest plot of unstandardized beta coefficients and 95% confidence intervals for the multilevel model effects of all videos on confidence in detecting relevant manipulation for the three respective manipulation tactics.** This shows that the long and short discrediting videos significantly increased confidence in detecting manipulative discrediting content compared to the control group. The long, but not the short scapegoating video significantly increased confidence in detecting manipulative scapegoating content compared to the control group. The long and short decontextualization videos significantly reduced confidence in detecting manipulative decontextualization content compared to the control group. *Note.* $N = 16,156$.

manipulation was significantly moderated by general manipulation discernment ability, $b = -0.04$, $SE = 0.02$, $t(16,130) = 2.08$, $p = .038$, such that the effect was only negative and significant at higher, but not at more moderate or lower levels of discernment (see Supplement, Section 2, Fig. S79). There was also a significant interaction between the long decontextualization video (vs. control) and voting behavior, $b = 0.14$, $SE = 0.07$, $t(16,120) = 1.97$, $p = .049$, whereby the effect was only negative and significant among those who voted in the June 2024 EU elections, but not among non-voters (Fig. S84). The effect of the long video was further moderated by longitude, $b = 0.01$, $SE = 0.01$, $t(16,130) = 2.52$, $p = .012$, being positive and significant among more Western and central European nations, but not Eastern ones (Fig. S80). Both the long, $b = -1.18$, $SE = 0.49$, $t(16,130) = 2.43$, $p = .010$, and short, $b = -4.27$, $SE = 1.99$, $t(15,330) = 2.15$, $p = .032$, videos were significantly moderated by education indices: the long video's positive effect was significant only in nations with moderate or higher education indices, while the short video's effect was non-significant across all levels (Fig. S81). The long video was also moderated by GDP per capita, $b = -0.01$, $SE = 0.01$, $t(16,130) = 4.40$, $p < .001$, and democratic indices, $b = -0.10$, $SE = 0.03$, $t(16,130) = 3.10$, $p = .002$, with effects significant only among nations with moderate or higher GDP per capita and democratic indices. The same pattern emerged for the short video, $b = -0.24$, $SE = 0.12$, $t(15,790) = 2.04$, $p = .041$ (Figs. S82–S83).

**Discrediting.** The positive effects of the long, $t(15,840) = 4.67$, $p < .001$, $d = 0.11$, 95% CI [0.06, 0.15], and short discrediting videos (vs. control), $t(15,700) = 3.68$, $p < .001$, $d = 0.15$, 95% CI [0.07, 0.24], on confidence in

detecting relevant manipulation were significant (see Fig. 3). There was also a significant interaction between the long discrediting video (vs. control) and age when predicting confidence in detecting manipulation, $b = 0.01$, SE $= 0.01$, $t(16,130) = 2.07$, $p = .039$, such that the effect was only positive and significant among older participants and participants aged closer to the mean, but not among younger participants (see Supplement, Section 2, Fig. S85).

## Sharing intentions
### Scapegoating
Sharing decisions. The positive effects of the long, $t(17,530) = 3.15$, $p = .002$, $d = 0.07$, 95% CI [0.03, 0.12], and short scapegoating videos (vs. control), $t(12,190) = 2.31$, $p = .021$, $d = 0.08$, 95% CI [0.01, 0.15], on sharing decisions for the scapegoating content were significant (see Fig. 4; sensitivity power analysis indicated that the effect size magnitude for the effect of the long scapegoating video (vs. control) on sharing discernment for the scapegoating content ($d = 0.08$) did not achieve the minimum effect size magnitude to reach a power of 0.80 ($d = 0.08$)). The positive effect of the long scapegoating video (vs. control) on sharing decisions for the scapegoating content was significantly moderated by general manipulation discernment ability, $b = 0.04$, $SE = 0.02$, $t(17,530) = 2.64$, $p = .008$, such that the effect was only positive and significant at higher and more moderate levels of general manipulation discernment ability, but non-significant at lower levels (see Supplement, Section 2, Fig. S25). Furthermore, the positive effect of the long scapegoating video (vs. control) on sharing decisions for the scapegoating content was significantly moderated by both digital literacy, $b = 0.08$, $SE = 0.03$, $t(17,350) = 2.30$, $p = .022$, and educational attainment, $b = 0.06$,

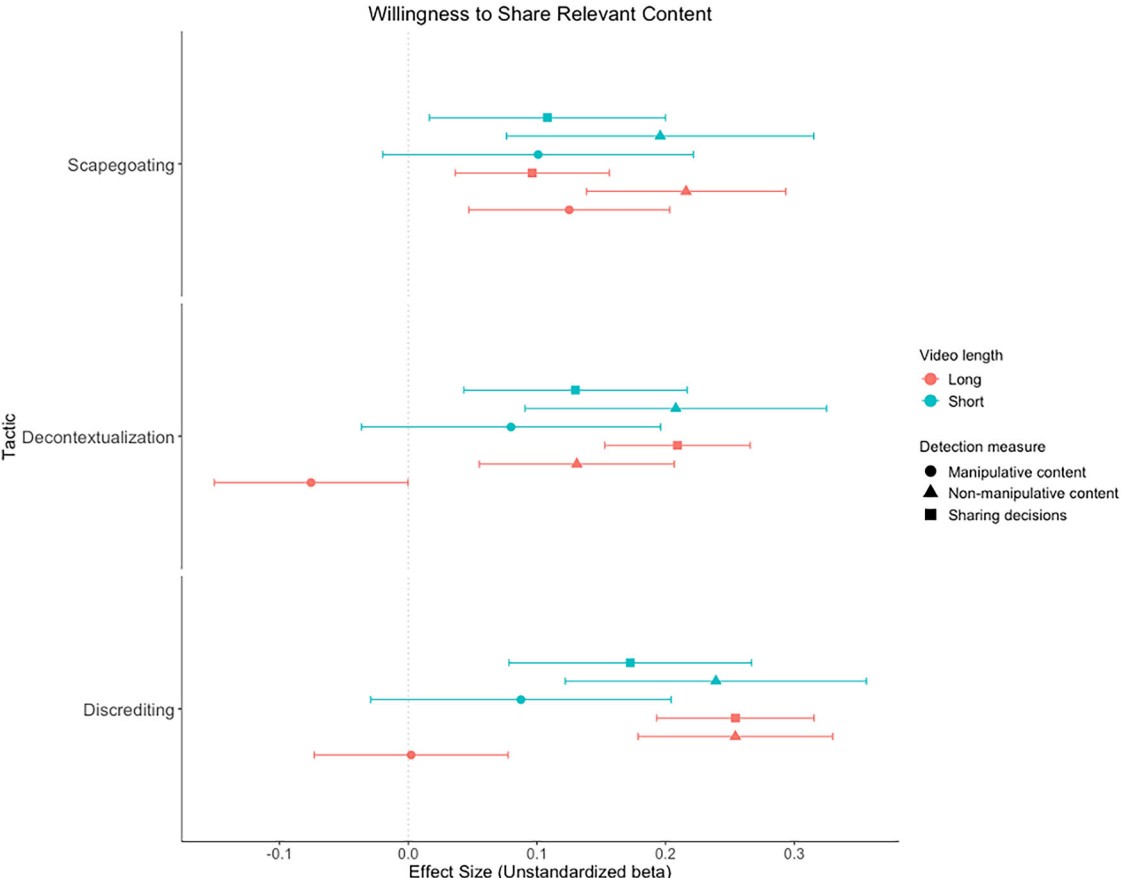

**Willingness to Share Relevant Content**

**Fig. 4 | Forest plot of unstandardized beta coefficients and 95% confidence intervals for the multilevel model effects of all videos on relevant sharing decisions and willingness to share relevant manipulative and non-manipulative content for the three respective manipulation tactics.** This shows that all of the long and short videos significantly increased relevant sharing decisions and willingness to share relevant non-manipulative content compared to the control group, while effects on willingness to share relevant manipulative content were mostly non-significant, the long scapegoating video significantly increased willingness to share manipulative scapegoating content, and the short discrediting video significantly increased willingness to share manipulative discrediting content compared to the control group. *Note. N* = 17,551 to 18,659. Positive effect sizes denote higher willingness to share manipulative and non-manipulative content, and higher willingness to share non-manipulative content relative to manipulative content for sharing decisions in the experimental conditions compared to the control group.

SE = 0.03, $t(17,530) = 2.25$, $p = .025$, such that the effect was only positive and significant at higher levels of digital literacy and educational attainment, but non-significant at more moderate and lower levels (see Supplement, Section 2, Figs. S26, S28). The positive effect of the short scapegoating video (vs. control) on sharing decisions for the scapegoating content was also significantly moderated by political ideology, $b = -0.06$, SE = 0.03, $t(17,530) = 1.98$, $p = .048$, such that the effect was only positive and significant among more politically left-wing participants, but not among more politically centrist or right-wing participants (see Supplement, Section 2, Fig. S27). The positive effect of the long scapegoating video (vs. control) on sharing decisions for the scapegoating content was also moderated by intentions to share the video within one's social network, $b = -0.14$, SE = 0.02, $t(18,610) = -6.91$, $p < .001$, such that the effect was only positive and significant at lower and more moderate levels of intentions to share, but not when intentions were higher (see Supplement, Section 2, Fig. S29).

Manipulative content. The long scapegoating video (vs. control) significantly increased willingness to share manipulative scapegoating content, $t(18,270) = 3.14$, $p = .002$, $d = 0.07$, 95% CI [0.03, 0.12] (see Fig. 4; sensitivity power analysis indicated that the effect size magnitude for the effect of the long scapegoating video (vs. control) on willingness to share manipulative scapegoating content ($d = 0.07$) did not achieve the minimum effect size magnitude to reach a power of 0.80 ($d = 0.08$)). In contrast, the short scapegoating video (vs. control) did not significantly alter willingness to share

manipulative scapegoating content, $t(18,250) = 1.64$, $p = .101$, $d = 0.06$, 95% CI [−0.01, 0.13] (the Bayes Factor for this effect indicated only anecdotal evidence for the alternative hypothesis over the null, $BF_{01} = 0.55$, $BF_{10} = 1.83$; see Fig. 4). The long and short scapegoating videos (vs. control) also significantly increased willingness to share manipulative discrediting content (see Supplement, Section 1).

Non-manipulative content. The long, $t(18,640) = 5.50$, $p < .001$, $d = 0.12$, 95% CI [0.08, 0.17], and short scapegoating videos (vs. control), $t(18,620) = 3.20$, $p = .001$, $d = 0.11$, 95% CI [0.04, 0.18], significantly increased willingness to share non-manipulative scapegoating content (see Fig. 4).

**Decontextualization**
Sharing decisions. The positive effects of the long, $t(17,990) = 7.26$, $p < .001$, $d = 0.16$, 95% CI [0.12, 0.21], and short decontextualization videos (vs. control), $t(13,900) = 2.93$, $p = .003$, $d = 0.10$, 95% CI [0.03, 0.17], on sharing decisions for the decontextualization content were significant (see Fig. 4). The positive effect of the long decontextualization video (vs. control) on sharing decisions for the decontextualization content was also moderated by intentions to share the video, $b = 0.07$, SE = 0.01, $t(17,970) = 4.79$, $p < .001$, such that the effect was only positive and significant when intentions to share were higher or more moderate, but not when intentions were lower (see Supplement, Section 2, Fig. S49).

Manipulative content. The long decontextualization video (vs. control) significantly reduced willingness to share manipulative decontextualization content, $t(18,720) = 1.97$, $p = .049$, $d = -0.04$, 95% CI [$-0.09$, $-0.01$], but the effect of the short decontextualization video (vs. control) was non-significant, $t(18,670) = 1.34$, $p = .179$, $d = 0.05$, 95% CI [$-0.02$, 0.12] (the Bayes Factor for the long decontextualization video (vs. control) indicated moderate evidence for the null hypothesis over the alternative, $BF_{01} = 5.4$, $BF_{10} = 0.18$, and the Bayes Factor for the short decontextualization video (vs. control) indicated only anecdotal evidence for the alternative hypothesis over the null, $BF_{01} = 0.52$, $BF_{10} = 1.92$; see Fig. 4). Sensitivity power analysis indicated that the effect size magnitude for the effect of the long decontextualization video (vs. control) on willingness to share the manipulative decontextualization content ($d = -0.04$) did not achieve the minimum effect size magnitude to reach a power of 0.80 ($d = 0.08$).

Non-manipulative content. Both the long, $t(18,720) = 3.40$, $p < .001$, $d = 0.08$, 95% CI [0.03, 0.12], and short decontextualization videos (vs. control), $t(18,710) = 3.58$, $p < .001$, $d = 0.12$, 95% CI [0.06, 0.19], significantly increased willingness to share the non-manipulative decontextualization content (see Fig. 4). The long decontextualization video (vs. control) also significantly reduced willingness to share the non-manipulative scapegoating content (see Supplement, Section 1).

### Discrediting
Sharing decisions. The positive effects of the long, $t(17,880) = 8.14$, $p < .001$, $d = 0.18$, 95% CI [0.14, 0.23], and short discrediting videos (vs. control), $t(13,350) = 3.58$, $p < .001$, $d = 0.13$, 95% CI [0.06, 0.19], on sharing decisions for the discrediting content were significant (see Fig. 4). The long discrediting video (vs. control) also significantly increased sharing decisions of the scapegoating and decontextualization content (see Supplement, Section 1). The positive effect of the long discrediting video (vs. control) on sharing decisions for the discrediting content was significantly moderated by general manipulation discernment ability, $b = 0.04$, SE = 0.02, $t(17,490) = 2.42$, $p = .015$, such that the effect was positive and significant at all levels of general manipulation discernment ability, but strongest at higher levels, weaker at more moderate levels, and weakest at lower levels (see Supplement, Section 2, Fig. S72). Furthermore, the positive effects of the long, $b = -0.06$, SE = 0.02, $t(17,490) = -2.73$, $p = .006$, and short discrediting videos (vs. control), $b = -0.08$, SE = 0.03, $t(17,490) = -2.37$, $p = .018$, on sharing decisions for the discrediting content were significantly moderated by political ideology, such that the effect for the long discrediting video (vs. control) was significant at all levels of political ideology, but strongest among more politically left-wing participants, weaker among more politically centrist participants, and weakest among more politically right-wing participants (see Supplement, Section 2, Fig. S73). The effect for the short discrediting video (vs. control) was only positive and significant among more politically left-wing and centrist participants, but not among more politically right-wing participants (see Supplement, Section 2, Fig. S73). The positive effect of the long discrediting video (vs. control) on sharing decisions for the discrediting content was also significantly moderated by gender, $t(17,430) = 2.70$, $p = .007$, such that the effect was positive and significant among both men and women, but it was strongest for women (see Supplement, Section 2, Fig. S75). The positive effect of the long discrediting video (vs. control) on sharing decisions for the discrediting content was moderated by intentions to share the video within one's social network, $b = 0.04$, SE = 0.02, $t(17,850) = 2.83$, $p = .005$, such that the effect was positive and significant when intentions to share were more moderate or higher, but was not significant when intentions to share were lower (see Supplement, Section 2, Fig. S76).

Manipulative content. The effects of the long, $t(18,660) = 0.06$, $p = .957$, $d = 0.01$, 95% CI [$-0.04$, 0.05], and short short discrediting videos (vs. control), $t(18,630) = 1.47$, $p = .142$, $d = 0.05$, 95% CI [$-0.02$, 0.12], on willingness to share the manipulative discrediting content were non-signficant (see Fig. 4). The Bayes Factor for the effect of the long discrediting

video (vs. control) indicated very strong evidence for the null hypothesis over the alternative, $BF_{01} = 39.3$, $BF_{10} = 0.03$, and the Bayes Factor for the effect of the short discrediting video (vs. control) indicated moderate evidence for the alternative hypothesis over the null, $BF_{01} = 0.31$, $BF_{10} = 3.25$.

Non-manipulative content. Furthermore, both the long, $t(18,640) = 6.56$, $p < .001$, $d = 0.15$, 95% CI [0.10, 0.19], and short discrediting videos (vs. control), $t(18,630) = 4.03$, $p < .001$, $d = 0.14$, 95% CI [0.07, 0.21], significantly increased willingness to share the non-manipulative discrediting content (see Fig. 4).

## Discussion
In this study, we tested the efficacy of three 50s prebunking videos (with three additional 20s versions) in improving manipulation discernment, sharing decisions, and confidence in detecting scapegoating, decontextualization, and discrediting content. Multilevel aggregate analysis of the 13 surveys across 12 EU nations indicated that all three of the long prebunking videos significantly improved manipulation discernment, technique discernment, and sharing decisions for relevant content compared to a control group (see Table 2 for a summary of results). Evidence also indicated that despite largely null effects on manipulation discernment and willingness to share the manipulative content for the shorter, 20s videos, these videos also improved technique recognition of the relevant manipulative content (although the effect of the short discrediting video did not achieve a minimum power of 0.80), sharing decisions for the relevant content (although the effect of the short scapegoating video did not achieve a minimum power of 0.80), and manipulation discernment for the short discrediting video (although the effect of the short discrediting video on manipulativeness assessments of manipulative discrediting content alone did not achieve a minimum power of 0.80). These findings replicate previous work demonstrating the effectiveness of shorter (e.g., 15s[9]) inoculation booster videos at improving the detection of relevant manipulation.

Extending Roozenbeek et al.'s[13] findings and replicating other similar work[14], we also found evidence of positive spill−overs or "cross-protection", wherein prebunking one manipulation tactic improved the detection of another, unrelated tactic. For example, the long decontextualization video significantly increased manipulation discernment of both the scapegoating and discrediting content compared to the control group, and each of the three long videos significantly increased technique recognition of the manipulative content for at least one other tactic (see Supplement, Section 1). Theoretically, this indicates that on top of straightforward educative effects (i.e., teaching participants what a specific tactic is and how to spot it), the inoculation videos tested here may also confer a more generalized resistance schema for discerning manipulative information from reliable content. As a result, we recommend this technique-based inoculation approach alongside more content-specific interventions such as fact checking due its scalability in reducing misinformation susceptibility across various domains[13]. Interestingly, although those who indicated having seen any of the videos before reported the highest confidence in detecting manipulation, those who indicated not having seen any of the videos previously were most likely to demonstrate the highest detection and discernment of relevant manipulation (see Supplement, Section 5). This suggests that psychological *booster shots* of prebunking messages are vital to provide individuals with the skills to detect misinformation that match assessments of their own abilities[9].

Overall, these findings suggest that although effect sizes were generally small, the 50s videos improved viewers' abilities to discern relevant manipulative content from non-manipulative content. Furthermore, even short 20s clips appeared to improve people's abilities to explicitly discern manipulation tactics from non-manipulative content. Both long and short content are thus likely to improve the reliability of information that people choose to interact with and share online. While the average effects of improved technique discernment here ($d = 0.08$) were generally small compared to other comparable lab work with similar designs on US samples ($d = 0.50$[13]), we suggest that this may be due to the heterogeneity in

**Table 2 | Summary of results and their implications for hypothesis testing**

| Dependent variable | Scapegoating | | Decontextualization | | Discrediting | |
|---|---|---|---|---|---|---|
| | **Long** | **Short** | **Long** | **Short** | **Long** | **Short** |
| **Manipulation discernment** | **Improved** | **Null** | **Improved** | **Null** | **Improved** | **Improved** |
| d [95% CI] | 0.08 [0.03, 0.12] | 0.03 [−0.04, 0.10] | 0.10 [0.06, 0.15] | 0.03 [−0.04, 0.10] | 0.12 [0.07, 0.16] | 0.09 [0.02, 0.16] |
| **Assessments (manipulative)** | **Improved** | **Null** | **Improved** | **Null** | **Improved** | **Improved** |
| d [95% CI] | 0.11 [0.07, 0.16] | 0.03 [−0.04, 0.10] | 0.11 [0.07, 0.16] | 0.06 [−0.01, 0.13] | 0.13 [0.08, 0.17] | 0.09 [0.02, 0.16] |
| **Assessments (non-manipulative)** | **Null** | **Null** | **Null** | **Null** | **Null** | **Null** |
| d [95% CI] | 0.02 [−0.02, 0.06] | 0.01 [−0.07, 0.07] | −0.01 [−0.05, 0.04] | 0.02 [−0.05, 0.09] | −0.03 [−0.07, 0.02] | −0.03 [−0.10, 0.40] |
| **Technique discernment** | **Improved** | **Improved** | **Improved** | **Improved** | **Improved** | **Improved** |
| d [95% CI] | 0.38 [0.34, 0.43] | 0.28 [0.21, 0.34] | 0.35 [0.30, 0.39] | 0.36 [.28, 0.42] | 0.20 [0.16, 0.25] | 0.14 [0.06, 0.20] |
| **Technique (manipulative)** | **Improved** | **Improved** | **Improved** | **Improved** | **Improved** | **Improved** |
| d [95% CI] | 0.36 [0.32, 0.41] | 0.21 [0.14, 0.27] | 0.32 [0.28, 0.36] | 0.28 [0.21, 0.35] | 0.21 [0.17, 0.25] | 0.10 [0.03, 0.17] |
| **Technique (non-manipulative)** | **Backfired** | **Backfired** | **Backfired** | **Backfired** | **Backfired** | **Backfired** |
| d [95% CI] | −0.20 [−0.24, −0.15] | −0.19 [−0.26, −0.13] | −0.19 [−0.23, −0.15] | −0.24 [−0.30, −0.17] | −0.13 [−0.17, −0.08] | −0.09 [−0.16, −0.02] |
| **Confidence** | **Improved** | **Null** | **Backfired** | **Backfired** | **Improved** | **Improved** |
| d [95% CI] | 0.11 [0.06, 0.18] | 0.07 [−0.01, 0.15] | −0.08 [−0.12, 0.03] | −0.07 [−0.15, 0.01] | 0.11 [0.06, 0.15] | 0.15 [0.07, 0.24] |
| **Sharing decisions** | **Improved** | **Improved** | **Improved** | **Improved** | **Improved** | **Improved** |
| d [95% CI] | 0.07 [0.03, 0.12] | 0.08 [0.01, 0.15] | 0.16 [0.12, 0.21] | 0.10 [0.03, 0.17] | 0.18 [0.14, 0.23] | 0.13 [0.06, 0.19] |
| **Sharing (manipulative)** | **Backfired** | **Null** | **Improved** | **Null** | **Null** | **Null** |
| d [95% CI] | 0.07 [0.03, 0.12] | 0.06 [−0.01, 0.13] | −0.04 [−0.09, −0.01] | 0.05 [−0.02, 0.12] | 0.01 [−0.04, 0.05] | 0.05 [−0.02, 0.12] |
| **Sharing (non-manipulative)** | **Improved** | **Improved** | **Improved** | **Improved** | **Improved** | **Improved** |
| d [95% CI] | 0.12 [0.08, 0.17] | 0.11 [0.04, 0.18] | 0.08 [0.03, 0.12] | 0.12 [0.06, 0.19] | 0.15 [0.10, 0.19] | 0.14 [0.07, 0.21] |

*Note.* Bold text denotes the nature of the effect. $d$ = Cohen's $d$ effect size; [95% CI] = 95% confidence interval for Cohen's $d$ effect size.

effectiveness of the videos across multiple EU nations when using aggregate analysis (e.g., for technique discernment of decontextualization content in Spain, $d = 0.28$, $p < .001$; but in Belgium (Flemish), $d = −0.16$, $p = .034$; see Supplement, Section 3).

Improvements in manipulation discernment were only indicated in Belgium, Germany, Hungary, Italy, and Poland, improvements in technique discernment were only indicated in Bulgaria, Germany, Hungary, the Netherlands, Romania, and Spain, and exclusively null effects for sharing discernment were indicated in Lithuania, the Netherlands, and Romania (see Supplement, Section 3). While some cross-cultural heterogeneity is expected, this mixed evidence coupled with the fact that all countries included in the current sample were European does raise an important question: Are we underestimating the cross-cultural variability in the efficacy of inoculation interventions[15]? Furthermore, our averaged effect size of $d = 0.08$ for manipulation discernment did not achieve the average power of 0.80 to detect an effect size of $d = 0.20$ in our samples, warranting caution when interpreting our findings. In sum, despite the promising significant aggregate effects, effect size magnitudes were notably small, warranting caution when expecting these interventions to work in the field[13]. Nevertheless, fieldwork testing the efficacy of these videos on over 1 million viewers garnered generally positive results[23], and similar fieldwork suggests that we can remain confident about the efficacy of prebunking videos for approximately 24h after viewing the content[13].

A number of interesting moderation effects also emerged, particularly with regard to country-level factors. For example, the scapegoating videos appeared to only improve technique discernment of the scapegoating content among participants from more Western or central European nations, and nations with moderate or higher education indices, GDP per capita, and democratic indices. A number of these may be attributed to

ceiling effects, such as the particularly high baseline discernment of scapegoating content indicated among participants from more Eastern European nations (see Supplement, Section 2). The conditional effects of the education indices of nations coupled with similar effects for personal educational attainment indicate that higher levels of education at both the country- and individual-level improve participants' receptivity to these prebunking interventions. Therefore, future research should focus on teaching general educational skills that improve individuals' critical thinking and provide them with the general ability to effectively appraise and evaluate the veracity of evidence outside of the prebunking context to ensure intervention effectiveness when implemented. In contrast, many of the videos appeared to be effective regardless of participants' general manipulation discernment abilities, age groups, levels of digital literacy, or voting behavior during the June 2024 EU elections. The lack of conditional effects of these variables on prebunking indicates that initial concerns over the role of these factors potentially dampening the effectiveness of misinformation interventions may be excessive.

At the same time, several variables—particularly political tolerance and political ideology—did appear to interact with the effectiveness of certain videos, suggesting that individuals who are already dispositionally open to alternative viewpoints may be more likely to benefit from prebunking. This aligns with past research on motivated reasoning and ideological asymmetry, in which higher political tolerance and openness are associated with a greater willingness to consider corrective or dissonant information[48]. For example, the scapegoating and discrediting videos were more effective among politically left-leaning or more politically tolerant participants, while effects were weaker or non-significant among more centrist or right-leaning groups. This pattern raises important questions about how to reach individuals who may be more ideologically rigid or resistant to interventions,

especially since these individuals may also be more vulnerable to manipulative content.

The interplay between individual-level and structural moderators also provides insight into the mechanisms through which prebunking works. That both personal educational attainment and national education indices moderated effectiveness suggests that interventions rely not only on individual cognitive capacity but also on broader educational infrastructure. These findings are consistent with dual-process models of reasoning, which highlight the role of both individual motivation and environmental cues in determining whether people engage in effortful evaluation[49]. In societies with stronger education systems or higher average educational attainment, people may be more habituated to critically appraising the intent behind persuasive messages, making them more receptive to prebunking messages.

Similarly, the moderating effects of GDP per capita and democratic indices suggest that the efficacy of these prebunking videos is shaped by broader political and economic conditions. The finding that some videos were more effective in more democratic or wealthier nations may reflect greater baseline media literacy, trust in institutions, or prior exposure to civic or critical thinking education. Conversely, in countries with very high or very low GDP or democracy scores, ceiling or floor effects may limit the observable impact of interventions. This indicates that while prebunking can be adapted across contexts, its success may hinge on tailoring the design or delivery to local structural conditions.

Interestingly, in contrast to some theoretical concerns, general manipulation discernment ability and digital literacy only occasionally moderated effects, and often in directions that favored already more discerning or digitally competent participants. While these results might suggest that people with stronger baseline abilities are more likely to benefit from prebunking, they also raise concerns about potentially widening informational inequalities: if such interventions are not made accessible and engaging for those with lower literacy or cognitive sophistication, the gap between the most and least resilient audiences may grow. Taken together, the moderation results underscore that although prebunking interventions can be broadly effective, they are not uniformly so. Rather, their impact is conditional on both individual-level characteristics (e.g., education, political tolerance) and contextual factors (e.g., education infrastructure). Recognizing these boundary conditions is essential for designing future interventions that are equitable, scalable, and sensitive to the populations they aim to support.

The effects of the videos on confidence in detecting manipulation appeared to depend on the content. While both discrediting videos and the long scapegoating video improved confidence compared to the control group, the decontextualization videos reduced confidence in detecting manipulation. This may be due to the nature of the tactics themselves. For example, participants may find it easier to detect unfair criticisms due to the consistent component in all discrediting manipulation of attempting to undermine someone's legitimacy. In contrast, decontextualization can take many forms, manipulating the context of images, quotes, or arguments in many different ways. Other research has reported null effects of a conspiracy theory prebunking game on confidence in detecting conspiracy theories[50], warranting further investigation into whether confidence is a necessary factor in providing people with the tools to detect manipulation.

While the videos tended to improve manipulation discernment through higher manipulativeness ratings of manipulative content and manipulativeness assessments of non-manipulative content remaining unchanged compared to the control group, technique recognition of non-manipulative content was actually slightly *reduced* by the prebunking videos compared to the control group. Furthermore, overall performance on this task was relatively poor. Roozenbeek and colleagues[13] used a similar design to our own, testing the effects of inoculation videos on the technique recognition of relevant manipulative and non-manipulative content. While the videos in Roozenbeek et al.'s[13] work did not alter technique recognition of the relevant non-manipulative content in most cases, the Ad Hominem video did significantly reduce technique recognition of the non-manipulative Ad Hominem content. We suggest that this may be because

the multiple-choice measures of technique recognition in our studies always included two decoy items (e.g., "Appeal to conflict"; see Supplement, Section 4), potentially introducing confounding uncertainty to participants through their deliberation on concepts that had not been introduced to them in the learning phase. Notable heterogeneity in significant improvements of discernment was also discovered when surveys were analyzed separately (see Supplement, Section 3). This is likely explained by the country-level factors that were found to interact with the effectiveness of the prebunking videos.

Alongside the reduced technique recognition of non-manipulative content, the long scapegoating video actually *increased* willingness to share relevant manipulative content, and only the long decontextualization video significantly reduced willingness to share relevant manipulative content. While our hypotheses on sharing intentions were exploratory, these findings have implications for the worsening of information hygiene in online spaces at first glance. However, the increased willingness to share non-manipulative content after watching all videos (long and short) translated into improved sharing discernment across all videos. Furthermore, the effects on sharing intentions appeared to be most pronounced among those who were already likely to share content online, as indicated by the moderation effects by intentions to share the videos within one's social network (see Supplement, Section 2). Taken together, these findings indicate that the inoculation videos presented here likely motivated participants' overall engagement with sharing any content online. While this risked higher intentions to share manipulative content, their overall decisions to share reliable content over manipulative content were still improved.

These results speak to a concern in the wider interventions literature: that interventions, while generally helpful, may occasionally lead to adverse or unintended effects—a phenomenon often described as "backfire"[51,52]. In our study, such backfire effects were observed in several forms, including reduced technique recognition of non-manipulative content, lower confidence in manipulation detection (especially for decontextualization), and increased willingness to share certain manipulative content. While these effects were typically small and context-dependent, they raise questions about the limits of prebunking interventions and the mechanisms through which they may inadvertently disrupt rather than reinforce accurate discernment. One possibility is that the increased cognitive load imposed by short, theory-rich videos prompts overgeneralization: participants may become hyper-vigilant to signs of manipulation and mistakenly apply learned labels to benign content. Alternatively, introducing manipulative tactics without sufficient scaffolding may reduce perceived self-efficacy or create doubt, especially when techniques are abstract or unfamiliar (as with decontextualization). These dynamics echo broader debates about psychological reactance, miscalibrated confidence, and the risk of cognitive fatigue when interventions are not carefully designed[53].

Another alternative explanation mirrors early findings in the debunking literature, where initial concerns about "backfire" effects turned out to be mostly due to methodological artefacts such as study design issues, item effects, low reliability of certain items, low power, and sample limitations. Larger trials revealed that such concerns about backfire following debunking were relatively rare, inconsistent, and did not replicate well[52,54,55]. In our case, internal consistency for the non-manipulative item sets was only modest ($\alpha \approx 0.58$–$0.70$), meaning that apparent effects on these items should be interpreted cautiously. As Swire-Thompson et al.[55] noted, low-reliability measures can inflate noise and make spurious patterns that resemble backfire effects more likely, particularly when true effects are small. Furthermore, although analysis of technique skepticism indicated that all videos increased skepticism, the effects of the videos on skepticism measured by the continuous manipulativeness assessments were all non-significant (see Supplement, Section 9). Future studies should thus focus on disentangling the reasons behind the notable differences between the effects of inoculation on technique recognition and manipulativeness assessment measures.

Attention should also be drawn to the *positive* unintended intervention effects discovered here, such as the cross-protection of prebunking a manipulation tactic, which, in some cases, conferred resistance against other

manipulation tactics. The small and inconsistent "backfire" findings reported here—which were generally weaker than the effects supporting our hypotheses—should be investigated using larger samples with high-reliability items. Overall, while we remain cautiously optimistic about the promise of prebunking, our findings do suggest that intervention designers should consider not just what is taught but how it is processed—balancing clarity, brevity, and conceptual differentiation to avoid confusing or demotivating participants.

The results also offer important insights into the discernment capacities of older adults. Although prior work has found that older individuals tend to be more susceptible to misinformation due to factors such as reduced cognitive flexibility, decreased memory performance, or increased reliance on heuristics[17,30], our findings suggest that older adults are nonetheless capable of benefiting from brief prebunking interventions. Across the 12 EU countries sampled, we observed improvements in manipulation and technique discernment, as well as in sharing decisions, despite the fact that all participants were aged 45 or older. This challenges narratives that position older adults as uniformly vulnerable and instead points to the potential of well-designed interventions to support discernment in this age group. While effects were smaller than in comparable studies with younger samples[13,21], our findings indicate that older populations can still meaningfully engage with and apply the lessons of inoculation interventions, particularly when videos are clear, brief, and accessible.

## Limitations
Despite the large samples and the varied number of nations included in the current research, there are a number of limitations that are still important to address. All videos were translated into native languages by native speakers with standardized terms used for the manipulation tactics. While this ensured consistency and interpretability across the surveys, the Anglicization of manipulation tactic terms in some contexts (e.g., *discréditation* for *discrediting* in France) may have introduced confounds into the vocabulary taught to participants in some cases. Furthermore, we intended to replicate the effectiveness of prebunking interventions among older populations simply to determine whether they are indeed effective among this understudied population. The effects appeared to replicate previous research on younger samples[7]. This suggests that the higher exposure and susceptibility to misinformation among older adults[17] does not act as a barrier to their receptivity to inoculation interventions. However, the absence of a younger sample meant that we were unable to directly compare the effects of these interventions across older and younger age groups, warranting further investigation from future work. Interestingly, effects were generally weaker than a study of the same intervention videos conducted on all age groups in a Hungarian sample[21], indicating that younger populations may indeed be more receptive to these interventions despite effectiveness across age groups.

Unlike previous research, which found a minimal confounding influence of the measurement scale used, item sentiments, and order effects in the piloting stage[13], the claims used in our item rating task were not formally pilot tested on perceived manipulativeness or other potentially confounding dimensions (e.g., emotional tone, complexity). Although each non-manipulative item was designed to mirror its manipulative counterpart in content and length, it is possible that unintended differences may have influenced participant ratings. To validate the categorization of manipulative and non-manipulative claims, we conducted paired samples t-tests on participants' perceived manipulativeness ratings. As intended by the authors, manipulative claims were rated as significantly more manipulative than non-manipulative claims in the control group alone. These analyses confirmed the same pattern across all three manipulation tactics: claims in the scapegoating, decontextualization, and discrediting measures that were labeled as manipulative were perceived as significantly more manipulative than non-manipulative claims (all $ps < 0.001$), supporting the robustness of our assumed categorization under baseline conditions. Furthermore, we ran exploratory multilevel models accounting for individual item variation (see Supplement, Section 6). In all cases, effects remained similar to the main analyses, and the vast majority of variance was accounted for by individual

differences between participants, rather than item variation. Nevertheless, we acknowledge that participants' interpretations of claims—especially in the absence of full context—may vary. For example, some may view a celebrity making a fan cry as sincere rather than manipulative, or consider a neutral-sounding statement like "researchers work hard to receive funding" as an oversimplification. While our classifications were statistically confirmed in the control group, this interpretive variability may contribute to unexplained variance in discernment scores and reflects broader challenges in operationalizing manipulation in brief, especially in cross-national research.

The use of a controlled experimental lab design provided us with the ability to ensure that participants watched and understood the content in the videos. In the real world, viewers are unlikely to pay attention to or comprehend every piece of information included in the videos. The effectiveness of the short videos at improving technique recognition of all relevant manipulative content suggests that these may be the most effective option for a paid media campaign, since they can play as unskippable adverts on YouTube to ensure that the content is attended to as much as possible. The manipulative and non-manipulative materials used in the item rating task were also experimenter-generated rather than sampled directly from real-world misinformation. Although this approach allowed us to tightly control the presence of specific manipulation tactics and to ensure comparability across languages and nations, it also introduces potential biases in ecological validity. The items may not fully capture the linguistic, emotional, or contextual nuances of naturally occurring misinformation. Future studies could complement this controlled design by testing prebunking effects on authentic, user-generated misinformation to examine the generalizability of our findings to real-world settings.

This study was conducted in the immediate aftermath of the June 2024 EU elections, a period marked by heightened exposure to manipulative political content. The three targeted tactics—scapegoating, decontextualization, and discrediting—were selected precisely because of their widespread use in election campaigns and political misinformation. That the videos improved discernment and sharing decisions for content reflecting these tactics suggests that prebunking may be a particularly timely and relevant tool in democratic contexts, where electoral discourse is vulnerable to manipulation. These findings support the use of prebunking as a scalable intervention during election cycles to help inoculate the public against politically motivated misinformation.

## Conclusion
This work demonstrates that although prebunking videos can improve manipulation discernment and sharing decisions for relevant content among older populations across 12 EU nations, effects are small and heterogeneous. This heterogeneity can be partially explained by conditional factors that reduce the efficacy of this intervention (i.e., less educated individuals in nations with lower education indices). Therefore, we recommend that future research replicates the effectiveness of prebunking videos further afield to non-EU nations and develops scalable ways to build adjacent skills that improve prebunking receptivity (e.g., through other misinformation-oriented education and skills feedback[56]).

## Data availability
All data, analysis code, materials, and stimuli used in this study are openly available on the Open Science Framework (OSF) at: https://osf.io/tkymv/overview?view_only=7486ede8ede1418f9a95855e7da5e5ac. This repository includes the anonymized datasets underlying all statistical analyses and figures, the full set of experimental materials (including video stimuli, survey instruments, and codebooks), and supplementary results. No restrictions apply to data access. All analysis code used to clean, process, and analyze the data, as well as the experimental scripts used to administer the surveys, are available in the same OSF repository as the data. Analyses were conducted in R (version 4.4.1) using packages including *lme4*, *emmeans*, *BayesFactor*, *dplyr*, *tidyverse*, and *ggplot2*. The version-controlled R scripts are included in the repository, with detailed annotations specifying model parameters,

variable processing, and figure generation procedures. No restrictions apply to code access. The pre-registration, including the analysis plan and hypotheses, is also available at https://osf.io/kjbdv/. All data supporting the findings of this study—including raw survey responses, cleaned analysis datasets, item stimuli, and all materials required to reproduce the analyses—are publicly available on the Open Science Framework (OSF) at: https://osf.io/tkymv/overview?view_only=7486ede8ede1418f9a95855e7da5e5ac. There are no access restrictions. This repository contains the minimum dataset necessary to interpret, verify, and extend the results reported in this article.

## Code availability

All custom analysis code used in this study—including data cleaning scripts, multilevel modeling code, Bayesian analyses, and figure-generation scripts—has been archived and is publicly accessible at https://osf.io/tkymv/overview?view_only=7486ede8ede1418f9a95855e7da5e5ac. The code is released under an open-source MIT licence and may be reused without restriction. This research was supported by Google Jigsaw.

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

## Acknowledgements

We would like to thank the local translators, data collection teams working with Bilendi, and related survey administrators across all participating EU nations for their invaluable contributions to adapting and implementing the materials in culturally and linguistically appropriate ways. We are also grateful to the Moonshot and Google Jigsaw teams for developing and distributing the prebunking videos tested in this research, and for their logistical support in coordinating the project. This research was supported by Google Jigsaw. The authors thank all study participants for their time and engagement, as well as colleagues at the University of Kent, University of Cambridge, and Moonshot for their feedback during project development.

## Author contributions

Mikey Biddlestone – Conceptualization, Methodology, Formal analysis, Investigation, Data curation, Software, Validation, Resources, Project administration, Visualization, Writing – original draft, Writing – review & editing. Beth Goldberg – Conceptualization, Methodology, Investigation, Resources, Project administration, Supervision, Visualization, Writing – review & editing, Funding acquisition. Melisa Basol – Conceptualization, Resources, Project administration, Visualization, Writing – review & editing. Katie Washington – Conceptualization, Investigation, Resources, Project administration. Sara Elnusairi – Conceptualization, Investigation, Resources, Project administration, Writing – review & editing, Funding acquisition. Anneka Sharpley – Conceptualization, Resources. Meghan Graham – Conceptualization, Investigation, Resources, Project administration, Writing – review & editing, Funding acquisition. Sander van der Linden – Methodology, Validation, Writing – review & editing. Ricky Green – Formal analysis, Validation, Visualization, Writing – review & editing. Jon Roozenbeek – Methodology, Validation, Writing – review & editing. Rachel Xu – Conceptualization, Methodology, Validation, Writing – review & editing. Andrew Pel – Conceptualization, Methodology, Investigation, Data curation, Validation, Resources, Project administration, Visualization, Writing – original draft, Writing – review & editing, Funding acquisition.

## Competing interests

The authors declare the following competing interests: Mikey Biddlestone was employed as a freelance researcher by Moonshot in a research position supervised by Andrew Pel. Moonshot and Google Jigsaw had already developed and distributed the prebunking video content, and tested its efficacy in field studies on YouTube. To receive a clearer and more nuanced picture of the efficacy of these interventions, Andrew Pel supervised a project in which Mikey Biddlestone tested the efficacy of the prebunking videos in the lab studies reported here. Mikey Biddlestone coordinated and planned the studies and data collection, with suggestions and conceptualization input from the Moonshot and Google Jigsaw team members. Importantly, none of the Moonshot or Google employees placed any restrictions on the design, data collection and analysis, decisions to publish or preparation of the manuscript, beyond the requirement that this work was to be done in compliance with their data policies and to better understand the efficacy of the prebunking videos (i.e., through selection of potential moderators and survey questions). Although the seventh author

now works as an independent, they were employed by Google at the time this data was collected and work was carried out, with the same role and input as all other Google employees. The other authors declare no competing interests.
