## [Transparent Peer Review file · Communications Psychology]

Video inoculation against election misinformation across 12 EU nations

Corresponding Author: Dr Mikey Biddlestone

Version 0:

Decision Letter:

Dear Dr Biddlestone,

Thank you for your patience during the peer-review process. Your manuscript titled "Video inoculation against election misinformation across 12 EU nations" has now been seen by 3 reviewers, and I include their comments at the end of this message. They find your work of interest but raised some important points. We are interested in the possibility of publishing your study in Communications Psychology, but would like to consider your responses to these concerns and assess a revised manuscript before we make a final decision on publication.

We therefore invite you to revise and resubmit your manuscript, along with a point-by-point response to the reviewers. Please highlight all changes in the manuscript text file.

Editorially, we consider it crucial that the claims of the study are well supported by the empirical results and not overstated in the revised manuscript. Per Reviewer #1's suggestion, results that do not support the main claims should also be discussed with appropriate statistical support. The potential confounds, such as those suggested by Reviewer #2, should be ruled out. Please also include additional reliability measures to test for internal consistency.

Please ensure you follow our statistical guidelines when reporting statistics (<https://www.nature.com/commpsychol/submit/submission-guidelines#statistical-guidelines>). Please note in particular our requirements for the reporting and interpretation of null-results. Non-significant findings derived from null-hypotheses significance tests should be reported in full, but may not be interpreted. Where you interpret null results, this interpretation must be based on Bayes Factors or equivalence tests.

We welcome the existence of preregistration and ask you to ensure that your revision complies with our respective guidelines to facilitate future steps. In particular, please ensure all deviations from the preregistration are reported. All originally preregistered hypotheses and analyses should be included, unless scientifically unsound, in which case the deviation needs to be highlighted and explained. Additional (exploratory) analyses may be included, but need to be labelled as post-hoc, non-preregistered. The full policy is here: <https://www.nature.com/commpsychol/editorial-policies/preregistration-policy>

I am attaching an Editorial Requests Table that details critical reporting requirements for the revised manuscript. Please attend to each item and ensure your manuscript is fully compliant. In particular, please make sure that the paper complies with the journal's formatting requirements as per the attached checklist, and that the paper does not include policy recommendations. If your revised manuscript is not aligned with these requests on major issues, such as those concerning statistics, it may be returned to you for further revisions without re-review.

Please submit the following items:

- Revised manuscript
- Point-by-point response to the referees' comments
- Cover letter (as a separate document)
- [Nature Research Reporting Summary](https://www.nature.com/documents/nr-reporting-summary.pdf)
- Completed Editorial Request Table (attached).

via this link: Link Redacted .

Additional guidance is available in our style and formatting guide [Communications Psychology formatting guide](https://www.nature.com/documents/commpsychol-style-formatting-guide-accept.pdf).

Best regards,

Troy Lui

Troy Lui, PhD
Associate Editor
Communications Psychology

REVIEWER EXPERTISE:

Reviewer #1: misinformation, inoculation, political psychology

Reviewer #2: misinformation, political psychology

Reviewer #3: misinformation, political psychology

REVIEWER REPORTS:

Reviewer #1 (Remarks to the Author):

The authors present a series of studies on the effectiveness of inoculation videos that have been conducted across 12 EU nations. The inoculation videos addressed three different manipulation tactics that are used in election campaigns, namely scapegoating, decontextualization and discrediting. The effectiveness of short and long versions of the videos have been tested with samples of participants aged 45 and older.

The paper includes an impressive set of studies that provide valuable insights into the effectiveness of inoculation interventions online. The analyses are well conducted and the level of transparency on the statistical level is helpful and rich. The paper is well written and provides a comprehensive overview of the research that includes additional analyses that are available in the Supplementary materials.

The paper provides novel insights into the effectiveness of video-based inoculation online. Yet, I have a couple of questions and suggestions that might help to improve the current version of the manuscript.

- The choice of testing the interventions with age 45+ samples does not really convince. Why exactly should these samples be targeted and what can we learn from the current findings? These questions should at least be addressed in the discussion.
- I missed a clearer outline of how the main DVs are aggregated. 16 items are presented with the first eight pertained to the

manipulation tactic prebunked. Does this indicated that 4 items are manipulative and 4 non-manipulative? And the remaining 8 items focused on the other two strategies. Does this indicated that for each strategy 2 items are manipulative and 2 non-manipulative?

- In a related manner, the pure number of analyses in this paper are quite difficult to oversee. There are seven experimental conditions with a series of DVs (including systematic variation of manipulateness and relatedness of items) in 12 countries. The number of analyses is so high that the paper would really benefit from a communication strategy that allows for a better understanding of the robustness of analyses across DVs and countries. Maybe, the authors can provide such an overview in a visualization? Ideally, this visualization would allow the reader to better understand where there are robust findings and where there is heterogeneity in the statistical results.

- It would be good to have descriptives and correlational tables involving the DVs in the Supplementary Materials.

- I suggest to use the terms direct and indirect instead of explicit and implicit. The latter terms seem to imply assumptions about the reflectiveness of cognitive processes that might not apply in the way the measures are used.

- This paper potentially allows for increasing our understanding of the specificity of intervention effects and I believe that this perspective would substantially increase the scientific merit of the paper. In the Supplementary Materials, the authors report analyses on spillover effects of interventions on the identification of manipulative tactics that have not been prebunked. Some of these analyses indicate significant effects in a similar range as the effects on related outcomes. I suggest that the authors include this as an important finding in the paper and discuss the implications. How can these effects be explained? Does this challenge the theoretical assumptions for how the interventions work? Where do we find spillover effects and where not? This question of specificity vs. spillover effects seems to be important to discuss in the paper.

- In regard to cross-cultural variation, it seems important to note that the cultural heterogeneity in the selection of nations is limited since they are all European countries.

Reviewer #2 (Remarks to the Author):

Thank you for the opportunity to review the manuscript "Video inoculation against election misinformation across 12 EU nations" which assessed the effectiveness of short (50 and 20 second) video inoculations that targeting three different manipulation tactics (scapegoating, decontextualization, and discrediting) in a 45+ population. The scale of this study, with almost 20000 participants in 12 different nations, is admirable, and I believe the assessment of the effectiveness of misinformation inoculation across different populations is both timely and important.

However, I have several concerns regarding the current manuscript, which I outline below.

1. Primarily, I found the argued benefits of inoculation tactics to be overstated based on the current results, and there to be insufficient discussion on both (1) the results that did not support the hypotheses, and (2) when the intervention either primarily or exclusively shifted participants' response bias to be more conservative rather than improved discernment. Given the current study was run under idealistic conditions (i.e., experimental setting where ratings of information occurred immediately after exposure to the intervention), the size and inconsistency of the effects is, to me, somewhat damning. Indeed, even if there is just a small decay in the effectiveness of the intervention after a delay (which research suggests to be likely), it is difficult to see these short interventions being impactful in practice (especially given it is unlikely people would view a 50 second ad in practice). I do believe these findings are worth publishing, but this should be done in a way that more accurately reflects the strength and limits of the evidence.

2. Relatedly to the first point, in some conditions, the intervention appears to increase confidence in identifying manipulative tactics without improving actual accuracy. This seems like a negative or even potentially harmful outcome, yet it is not acknowledged as such in the discussion. The variability in effects across the three manipulation types and across populations further suggests that the intervention's impact is, at best, inconsistent. These results seem to challenge the commonly cited generalizability and scalability of inoculation strategies. I recommend that these limitations be more prominently addressed in the discussion, including more focus on which populations the intervention is—and is not—effective in.

3. As with similar studies in this area, I hold concerns about the appropriateness of claims used to assess the effectiveness of the intervention.

a. For one, being able to identify a potentially manipulative tactic is not equivalent to being able to discern true and false information, and I find the presentation of these findings as targeting misinformation to be misleading. Moreover, it is unclear to me how the paper is directly targeting election content, which is presented as one of the key aims of the piece. These factors draw into question the appropriateness of the design to target the key research question.

b. The decontextualized claims rely on the participants imagining being presented with a claim, rather than a claim actually being presented. This creates a confound where for some claims the participant could reasonably imagine the claim to be manipulative or non-manipulative, unrelated to how the authors have classed these claims. For example, a clip of a comedian making a fan cry could plausibly happen, and isn't necessarily manipulative, whilst a picture of a line of people with the caption about attending concerts despite cost of living could easily be an image taken out of context. Similarly, certain claims in the 'discrediting' condition (e.g., "researchers work hard to receive funding") could easily be classed as an oversimplification. These issues with the claims draws into question the ability to draw relevant inferences about participants ability to discern manipulative and non-manipulative content.

c. It is also unclear to me how these claims have been matched other than including somewhat similar content. For example, were these claims pilot tested on the important characteristics of interest? This could easily impact the inferences drawn - e.g., if 'non-manipulative' and 'manipulative' content are starting from very different points on measures of interest (e.g.,

ratings of manipulativeness) I worry that subtraction scores would be impacted by the scale used (indeed, I worry about the information lost by using subtraction scores more generally). I think the reader would also benefit from figures with absolute values (rather than unstandardized beta weights) as this is more intuitive to understand the size of the effects.

4. There appears to be some deviation from the pre-registered analysis plan (or at least it is unclear to me as a reader whether the analyses presented are the same as those pre-registered). I would recommend the authors outline any instance where the analyses presented deviate from the pre-registration, or more clearly state the analyses used.

5. The abstract appears to suggest that the 50 second videos are significantly more effective than the 20 second, however, the relative effectiveness of the different videos does not appear to be directly compared.

I wish the authors all the best with their manuscript.

Reviewer #3 (Remarks to the Author):

Thank you for the opportunity to review this manuscript. In general, I think this is an interesting topic and important study. I think the study was well-written. That said, I had some concerns about the theories motivating the study, the takeaways, and methodological decisions. I will first focus on larger comments and then provide a set of more minor comments.

MAJOR

1. Generally, I found the Introduction to be well-written, and I understand the authors are probably working with space constraints. That said, I found certain theoretical pieces to be lacking. In their absence, it was at times difficult to evaluate the novelty and importance of the study. I offer a handful of suggestions here to bolster the theoretical framework.

a. It would have been helpful to contextualize and set up the DVs in the Introduction. I was not quite prepared to interpret the Results. A subsection dedicated to explaining the study outcomes in the Introduction would be very helpful.

b. There are a multitude of theoretically relevant moderators. Why did the authors focus on political tolerance and digital literacy relative to other potential moderators? Note, I do not think the authors should have necessarily included other moderators, but I do think they need to explain and justify why these moderators were important to focus on.

c. Similarly, what did the authors expect for the WEIRDness, demographics, and general manipulation discernment moderation results? What do these analyses tell us? I wasn't sure how to interpret the moderation results without the necessary background.

2. Similarly, I think some crucial elements were missing from the Discussion. Again, in the absence of theoretical interpretation, it can be difficult to situate the study within the larger literature.

a. More real estate should be dedicated to walking through the moderation results. What do the smattering of significant results tell us about the constructs and the theoretical framework?

b. The authors did not really speak to the fact that participants were significantly more willing to share manipulative discrediting content after watching the short discrediting video relative to control. Discussion and interpretation for this effect are warranted.

c. I think the authors need to speak more in the Discussion to the potential backfire effects of prebunking in the study (lower confidence, more willingness to share manipulative content, reduced detection of non-manipulative content) and how that links to larger debates in the literature about prebunking. What do these results tell us about debates in the literature and how should we think about prebunking moving forward?

d. It seems that studying older adults was a primary motivation for the study, yet there was little discussion about what these results tell us about older adults. I think far more is needed in the Discussion regarding what these results show us about discernment in older adults. Similarly, the authors do not really return to the main point of focusing on elections either.

e. I think some of the recommendations (e.g., about particularly vulnerable populations) feel like a stretch considering the heterogeneity of the findings and the inconsistency of the significant moderation results. I encourage the authors to temper their conclusions and/or better justify their conclusions.

3. Finally, I had concerns about some of the methodological choices and internal consistency statistics.

a. The internal consistency statistics for the manipulativeness ratings for the non-manipulative decontextualization and discrediting statements were rather low. Can the authors speak to other metrics of reliability for these ratings? I worry about the reliability of the difference scores when the reliabilities of these ratings were quite low. Similarly, the reliability of the political tolerance scale was rather low.

b. Performance on the explicit detection task was rather poor (at least poorer than I might have expected). Can the authors elaborate on whether and to what extent these averages might have impacted the results?

c. General manipulation discernment was calculated as a sum score, implicit manipulation discernment was calculated as a proportion correct across both types of statements, and sharing discernment was calculated as a difference score. Why did the authors score discernment differently across the different DVs? This not only might affect the results but the interpretation of the results. I would encourage the authors to score discernment in a similar fashion across DVs or better justify why discernment is scored differently across DVs (and how these scoring differences affect the interpretation of what discernment means).

MINOR

4. The power analyses were for effects that were much larger than found for the implicit assessments, and the sensitivity analyses also indicate 80% power to detect effects that were larger than found in the study. I have no problem with the power analysis, as it was theoretically sound and based on prior research. I do think the authors need to speak to the power in their study, however, based on the effect sizes (more than they do currently). Can the authors do a sensitivity analysis based on the identified effect sizes, especially for implicit assessments? What was the likely power to detect those small effects? If the power was low, the authors should speak to that and interpret accordingly.

5. I understand this was a secondary analysis, but I am a bit puzzled by the results comparing between groups of participants who had seen, had not seen, or were unsure of whether they had seen the videos before. Why would there be differences between these groups? Was the finding expected? If so, why or why not? I currently do not have an understanding of why there would be these identified differences between the groups.
6. The confidence in detecting manipulation subsection of the Results did not have the same organization as prior subsections. Can the authors be consistent and present the results by scapegoating, discrediting, and decontextualization separately? It would go a long way for ease of interpretation.
7. I understand the results surrounding higher levels of education, but what does it mean to "teach general educational skills"? What does this recommendation specifically mean? That people should receive more education?

Version 1:

Decision Letter:

Dear Dr Biddlestone,

Your manuscript titled "Video inoculation against election misinformation across 12 EU nations" has now been seen by our reviewers, whose comments appear below. In light of their advice I am delighted to say that we are happy, in principle, to publish a suitably revised version in Communications Psychology.

We therefore invite you to revise your paper one last time to address the remaining concerns of our reviewers and a list of editorial requests. At the same time we ask that you edit your manuscript to comply with our format requirements and to maximise the accessibility and therefore the impact of your work.

EDITORIAL REQUESTS:

SUBMISSION INFORMATION:

OPEN ACCESS:

Communications Psychology is a fully open access journal. Articles are made freely accessible on publication. For further

information about article processing charges, open access funding, and advice and support from Nature Research, please visit <https://www.nature.com/commpsychol/open-access>

* **DATA AVAILABILITY:**

Link Redacted

Best regards,

Troy Lui

Troy Lui, PhD
Associate Editor
Communications Psychology

REVIEWERS' COMMENTS:

Reviewer #1 (Remarks to the Author):

I want to thank the authors for carefully considering my comments and suggestions. I have no further questions or concerns.

Reviewer #2 (Remarks to the Author):

Thank you for the opportunity to review the revised version of the manuscript "Video inoculation against election misinformation across 12 EU nations". I appreciate the authors' attempts to address my own and the other reviewers' concerns, I believe this has generally made the relative strength of the findings clearer. This said, I do remain concerned about a couple of factors, and have a few more minor recommendations that I believe would improve the flow of the paper.

Firstly, I understand that the authors pushed back on my initial point regarding the definition of "misinformation." I acknowledge that this is partly due to unresolved epistemological issues in the field (which are beyond the scope of the current piece), as well as a lack of clarity in my original review. My major concern, however, is with the inferences drawn from the materials. As I understand it (and please correct me if I am wrong), rather than real-world claims these materials were created by the authors to contain the manipulative tactic that is both (1) the target of the intervention and (2) presented as representative of real-world misinformation. I think it is important to explicitly acknowledge the potential biases/confounds that might be introduced by using experimenter created materials (i.e., it is unclear the extent these materials reflect those in

the real-world). A couple of sentences in the limitations section of the discussion should be sufficient for this.

Secondly, given its length and repetitiveness I found the results section somewhat laborious to read, and I think it would benefit from streamlining. Namely, for the reader I believe it would be beneficial to group results pertaining to manipulative discernment, technique recognition, and confidence together, given they are all directly related. As it reads currently, I found myself having to flick between sections to get a real understanding of the pattern of results (particularly when it came to the relationship between improved ability and improved confidence). Noting the interaction effects when discussing the main result, rather than having a separate section, I think would also aid readability. Please at least present the interactions sections in the same format across all results, as the switch in structure in the "Confidence in detecting manipulation" was not super intuitive. Finally, I think presenting the results for manipulative discernment prior to the results isolated to manipulative and non-manipulative content would be more intuitive for the reader, given it is similar in nature to breaking down an interaction.

Some smaller points:

The addition of table 1 is fantastic, however, please add the specific results here as well to allow for a quick understanding of effect size.

I think some care needs to be made with the claim that age "robustly predicts sharing" (pg. 6), as the Guess finding is relatively old and there is some more recent evidence that suggests this relationship is less clear cut (e.g., <https://doi.org/10.1007/s12144-023-04464-w>; <https://doi.org/10.1038/s41598-023-34402-6>). I think it would be beneficial to acknowledge that the Guess paper is not the only paper that has assessed this relationship.

Interpretation of BFs would be aided by reporting BF01 for null results (rather than BF10). Additionally, reporting BFs for marginal p values (e.g., .049) would also be useful to quantify the strength of evidence.

The framing of effect sizes "not reaching .8 power" is odd, especially given sensitivity analyses are dependent on the size of the effects observed. Discussing the size of the effects rather than the power to observe these effects is more beneficial, especially given how large the sample size is.

A number of times an interaction effect is noted, however, there is no description of what drives the interaction. Instead, it is simply stated that the effect is significant at all levels (pp. 20, 22 and 23). For example, on p.20: "The positive effects of both the long and short scapegoating videos (vs. control) on technique recognition of the manipulative scapegoating content were moderated by intentions to share the video, such that the effect of the long scapegoating video (vs. control) was positive and significant at higher, more moderate, and lower levels of intentions to share." Please describe what drives the interaction.

I think more care needs to be made with the recommendation for the short inoculation videos given a lot of these effects were null, particularly sharing of manipulative content. In a similar vein, given fact-checking and inoculation were not directly compared, how small the effect sizes observed were, and the fact that fact-checking and inoculation are not at odds with each other the recommendation for inoculation above fact checking (p.34) seems unnecessary. Indeed, it seems better to advocate for multiple interventions in conjunction, rather than any one above another.

I wish the authors all the best with this manuscript.

COMMSPSYCHOL-25-0251A

Dear Professor Lui,

Thank you for your message dated 3rd June 2025, regarding the above paper that we submitted to *Communications Psychology*. We appreciate your invitation to revise and resubmit the paper for further consideration and we are grateful for yours and the reviewers' generous guidance on how to improve the paper and highlight its unique and important contributions to the literature. Thank you also for allowing us an extension to revise the manuscript and we apologise for submitting the revision slightly late.

We feel that the paper is much improved as a result of this process, and in our revision letter, we outline in detail how we have addressed each of yours and the reviewers' comments.

Editor's comments

1) Per Reviewer #1's suggestion, results that do not support the main claims should also be discussed with appropriate statistical support.

Thank you for this important suggestion. To address this concern, we have now included a new table 1 in the discussion on p. X which summarises the results and their implications for our hypothesis testing.

We have also spent much more time interpreting these different findings in the discussion. For example, pp. 39-41 now reads:

“Alongside the reduced technique recognition of non-manipulative content, the long scapegoating and discrediting videos actually increased willingness to share relevant manipulative content (although the effect for the short discrediting video did not achieve a minimum power of .80), and all other videos had no effect on this content. While our hypotheses on sharing intentions were exploratory, these findings have implications for the worsening of information hygiene in online spaces at first glance. However, the increased willingness to share non-manipulative content after watching all videos (long and short) translated into improved sharing discernment across all videos. Furthermore, the effects on sharing intentions appeared to be most pronounced among those who were already likely to share content online, as indicated by the moderation effects by intentions to share the videos within one's social network (see Supplement, Section 2). Taken together, these findings indicate that the inoculation videos presented here likely motivated participants' overall engagement with sharing any content online. While this risked higher intentions to share manipulative content, their overall decisions to share reliable content over manipulative content were still improved.

These results speak to a concern in the wider interventions literature: that interventions, while generally helpful, may occasionally lead to adverse or unintended effects—a phenomenon often described as “backfire” (e.g., Hoes et al., 2024; Swire-Thompson et al., 2020). In our study, such backfire effects were observed in several forms, including reduced technique recognition of non-manipulative content, lower confidence in manipulation detection (especially for decontextualization), and increased willingness to share certain manipulative content. While these effects were typically small and context-dependent, they

raise questions about the limits of prebunking interventions and the mechanisms through which they may inadvertently disrupt rather than reinforce accurate discernment. One possibility is that the increased cognitive load imposed by short, theory-rich videos prompts overgeneralization: participants may become hyper-vigilant to signs of manipulation and mistakenly apply learned labels to benign content. Alternatively, introducing manipulative tactics without sufficient scaffolding may reduce perceived self-efficacy or create doubt, especially when techniques are abstract or unfamiliar (as with decontextualization). These dynamics echo broader debates about psychological reactance, miscalibrated confidence, and the risk of cognitive fatigue when interventions are not carefully designed (see Tang, 2025).

Another alternative explanation mirrors early findings in the debunking literature where initial concerns about “backfire” effects turned out to be mostly due to methodological artefacts such as study design issues, item effects, low reliability of certain items, low power, and sample limitations. Larger trials revealed that such concerns about backfire following debunking were relatively rare, inconsistent, and did not replicate well (Wood & Porter, 2019; Swire-Thompson et al., 2020; 2022). In our case, internal consistency for the non-manipulative item set was only modest ($\alpha \approx .58-.70$), meaning that apparent effects on these items should be interpreted cautiously. As Swire-Thompson and colleagues (2020) noted, low-reliability measures can inflate noise and make spurious patterns that resemble backfire effects more likely, particularly when true effects are small. Furthermore, although analysis of technique skepticism indicated that all videos increased skepticism, the effects of the videos on skepticism measured by the continuous manipulateness assessments were all non-significant (see Supplement, Section 9). Future studies should thus focus on disentangling the reasons behind the notable differences between the effects of inoculation on technique recognition and manipulateness assessment measures.

Attention should also be drawn to the positive unintended intervention effects discovered here, such as the cross-protection of prebunking a manipulation tactic, which, in some cases, conferred resistance against other manipulation tactics. The small and inconsistent “backfire” findings reported here—which were generally weaker than the effects supporting our hypotheses—should be investigated using larger samples with high-reliability items. Overall, while we remain cautiously optimistic about the promise of prebunking, our findings do suggest that intervention designers should consider not just what is taught but how it is processed—balancing clarity, brevity, and conceptual differentiation to avoid confusing or demotivating participants.”

2) The potential confounds, such as those suggested by Reviewer #2, should be ruled out.

We found this comment to be incredibly helpful in motivating us to run additional analyses to better understand the reliability of our items. Firstly, we explored paired samples t-tests showing that manipulative items were indeed rated as significantly more manipulative than non-manipulative items. Furthermore, we added exploratory multilevel models controlling for individual items (manipulative and non-manipulative) to assess the variance accounted for by item variation. These findings indicated that while there was some variance in a number of cases, the vast majority of variance could instead be attributed to individual differences between participants. Furthermore, results remained almost identical to our main analyses when item variation was taken into account, hopefully reducing concerns from yourself and the reviewers about the reliability of our item rating task.

The details of the robustness checks showing higher perceived manipulateness of the manipulative items can be found in footnotes in the main text, for example on pp. 13-14:

“As a robustness check, we conducted a paired samples t-test comparing manipulateness assessments of the manipulative vs. non-manipulative scapegoating content only on participants in the control group. The manipulative scapegoating content was perceived as significantly more manipulative, $M = 4.49$, $SD = 1.71$, than the non-manipulative scapegoating content, $M = 3.86$, $SD = 1.50$, $t(3,777) = 18.84$, $p < .001$, $d = 0.31$, validating this as a measure of manipulation discernment.”

Furthermore, the exploratory multilevel models accounting for individual item variation can be found in the Supplement in Section 6.

3) Please also include additional reliability measures to test for internal consistency.

We have now included reliability statistics for all relevant variables in our *Measures* subsection of the **Methods** section. We also hope that our exploratory multilevel models accounting for the variance in individual items further addresses understandable concerns regarding the reliability of our item rating task.

4) Please ensure you follow our statistical guidelines when reporting statistics (<https://www.nature.com/commspsychol/submit/submission-guidelines#statistical-guidelines>). Please note in particular our requirements for the reporting and interpretation of null-results. Non-significant findings derived from null-hypotheses significance tests should be reported in full, but may not be interpreted. Where you interpret null results, this interpretation must be based on Bayes Factors or equivalence tests.

We have now conducted Bayesian analysis for the null effects. The details of these can be found in footnotes in the relevant results section areas. We have also explained our addition of these analyses before the results section on p. 12:

“To evaluate the strength of evidence for null versus alternative hypotheses in cases of non-significant findings, we conducted Bayesian independent-samples t-tests using the default JZS prior (Cauchy scale $r = .71$) as implemented in the BayesFactor package in R. These analyses allowed us to quantify the degree of support for the null hypothesis, supplementing traditional p-values with continuous measures of evidence strength. The relevant contrasts were specified for each relevant experimental condition vs. the control group.”

5) We welcome the existence of preregistration and ask you to ensure that your revision complies with our respective guidelines to facilitate future steps. In particular, please ensure all deviations from the preregistration are reported. All originally preregistered hypotheses and analyses should be included, unless scientifically unsound, in which case the deviation needs to be highlighted and explained. Additional (exploratory) analyses may be included, but need to be labelled as post-hoc, non-preregistered. The full policy is here: <https://www.nature.com/commspsychol/editorial-policies/preregistration-policy>

We have now included a list of our directional vs. exploratory analyses outlined in our pre-registration document on pp. 9-11:

“Pre-registered directional hypotheses

H1) Watching a relevant prebunking video will significantly increase implicit manipulation discernment (i.e., rating manipulative content as more manipulative and non-manipulative content as less manipulative compared to the control group) for that tactic.

H2) Watching a relevant prebunking video will significantly increase explicit manipulation discernment (i.e., correctly identifying the relevant manipulation tactic for manipulative items and “None of the above” for non-manipulative items more frequently than the control group) for that tactic.

H3) Watching a relevant prebunking video will significantly increase implicit detection of manipulation (i.e., higher manipulateness ratings for manipulative items compared to the control group) for that tactic.

H4) Watching a relevant prebunking video will significantly increase explicit detection of manipulation (i.e., correctly identifying the relevant manipulation tactic for manipulative items more frequently than the control group) for that tactic.

H5) Watching a relevant prebunking video will significantly improve sharing decisions (i.e., lower willingness to share manipulative items and higher willingness to share non-manipulative items compared to the control group) for that tactic.

H6) Watching a relevant prebunking video will significantly decrease willingness to share manipulative content compared to the control group for that tactic.

Pre-registered exploratory research questions

RQ1) Will watching a relevant prebunking video significantly affect implicit detection of non-manipulative content (i.e., lower manipulateness ratings for non-manipulative items compared to the control group) for that tactic?

RQ2) Will watching a relevant prebunking video significantly affect explicit detection of non-manipulative content (i.e., more frequent correct “None of the above” responses for non-manipulative items than the control group) for that tactic?

RQ3) Will watching a relevant prebunking video significantly affect willingness to share non-manipulative content (i.e., higher willingness to share non-manipulative items compared to the control group) for that tactic?

RQ4) Will watching a relevant prebunking video significantly affect confidence in detecting manipulation compared to the control group for that tactic?

RQ5) Cross-protection: Will watching a prebunking video for one tactic improve implicit manipulation discernment, explicit manipulation discernment, and sharing decisions compared to the control group for other, non-prebunked tactics?

RQ6) Skepticism/distrust: Will any prebunking condition increase skepticism (i.e., rating both manipulative and non-manipulative content as more manipulative, or selecting a manipulation tactic for both manipulative and non-manipulative items more frequently than the control group)?

RQ7) Naïveté/gullibility: Will any prebunking condition increase naïveté (i.e., rating both manipulative and non-manipulative content as less manipulative, or selecting “None of the above” for both manipulative and non-manipulative items than the control group)?

RQ8) Will any hypothesized or exploratory effects be moderated by any of the following variables: general manipulation discernment, voting behaviour in the June EU elections, intentions to share videos within one’s social network, digital literacy, socio-demographics (age, gender, education, political ideology), or country-level indices (latitude, education index, industrialisation index, GDP per capita, democratic index)?

Pre-registration deviations

In our pre-registration document, we also planned to explore whether any of the effects were moderated by video length. Furthermore, we collected information on whether participants believed they had seen, not seen, or were uncertain whether they had seen the video in question before. While we were unable to include these variables as moderators because participants in the control group could not provide comparable information, we instead analyzed whether there were any significant differences in our main outcomes variables by video length and whether participants believed they had seen the videos before. The results for these analyses can be found in the Supplement (Sections 5 and 6).”

Reviewer 1’s comments

- 1) The paper includes an impressive set of studies that provide valuable insights into the effectiveness of inoculation interventions online. The analyses are well conducted and the level of transparency on the statistical level is helpful and rich. The paper is well written and provides a comprehensive overview of the research that includes additional analyses that are available in the Supplementary materials.**

Thank you for your supportive comments. We are glad you recognised the novel insights we hoped to provide for the literature.

- 2) The choice of testing the interventions with age 45+ samples does not really convince. Why exactly should these samples be targeted and what can we learn from the current findings? These questions should at least be addressed in the discussion.**

We appreciate the need to clarify this reasoning further. Firstly, please refer to our paragraph on p. 6 in which we initially explained this reasoning:

“Older populations are particularly vulnerable to political manipulation due to digital education exclusion (Tomczyk et al., 2023) and their heavy reliance on familiar sources such as news portals and social media (Bergh, 2023; but see Kyrychenko et al., 2024 for evidence of higher susceptibility to misinformation among younger people). They are also more likely to share misinformation (Guess et al., 2019) and, given their higher voter turnout (Goerres, 2007), represent a key consequential demographic for political inoculation efforts. Brashier and Schacter (2020) identify three factors that may influence misinformation susceptibility across age groups: cognitive decline, social change, and digital literacy. Cognitive decline can increase misinformation susceptibility by impairing memory for information sources and reinforcing reliance on familiar, repeated claims (Fazio et al., 2019). Social change may also play a role, as greater interpersonal trust (or reduced institutional trust) over the lifespan may make individuals more likely to believe misinformation from close or trusted sources, or less from official or governmental sources. Additionally, the declining cognitive component of digital literacy—critical thinking (see Ng, 2013)—over the lifespan is linked to weaker misinformation resistance (Eshet-Alkalai & Chujut, 2010; Gaillard et al., 2021). Finally, older adults are rarely in education, unlike people under 20, which complicates the implementation of educational or literacy programmes among this demographic. To promote digital inclusion among this important demographic and replicate previous inoculation effects among older age groups, we tested our inoculation videos among participants aged 45 and older.”

We have also added content to our discussion of the implications and limitations of our focus on age on pp. 1-2, which we hope clarifies this decision further:

“Furthermore, we intended to replicate the effectiveness of prebunking interventions among older populations simply to determine whether they are indeed effective among this understudied population. The effects appeared to replicate previous research on younger samples (e.g., Lu et al., 2023). This suggests that despite the higher exposure and susceptibility to misinformation among older adults (Guess et al., 2019) does not act as a barrier to their receptivity to inoculation interventions. However, the absence of a younger sample meant that we were unable to directly compare the effects of these interventions across older and younger age groups, warranting further investigation from future work. Interestingly, effects were generally weaker than a study of the same intervention videos conducted on all age groups in a Hungarian sample (Orosz et al., 2024), indicating that younger populations may indeed be more receptive to these interventions despite effectiveness across age groups.”

- 3) I missed a clearer outline of how the main DVs are aggregated. 16 items are presented with the first eight pertained to the manipulation tactic prebunked. Does this indicated that 4 items are manipulative and 4 non-manipulative? And the remaining 8 items focused on the other two strategies. Does this indicated that for each strategy 2 items are manipulative and 2 non-manipulative?**

Thank you for raising this lack of clarity. We have now elaborated on this in our methods section on pp. 8-9:

“The items were grouped by tactic and presented in randomized order. For each counterpart, participants were randomly presented with either the manipulative or non-manipulative counterpart. In the experimental conditions, participants were presented with a total of 16 items. The first eight pertained to the manipulation tactic prebunked in the video they viewed, and then they were presented with four items from each of the other two respective tactics grouped by tactic in randomized order. The number of manipulative and non-manipulative posts within each category was also completely randomized. For each item, participants rated their perceived manipulateness of the content (manipulateness assessments), their willingness to share the content within their social network (willingness to share), and a multiple choice list of four options (technique recognition). The measure of technique recognition always included the correct manipulation tactic (e.g., “Scapegoating”), two decoy answers, and “None of the above” to indicate no manipulation. The order of the respective manipulation tactic answer and two decoy answers were always randomized, while the “None of the above” option was always presented at the end.”

- 4) In a related manner, the pure number of analyses in this paper are quite difficult to oversee. There are seven experimental conditions with a series of DVs (including systematic variation of manipulateness and relatedness of items) in 12 countries. The number of analyses is so high that the paper would really benefit from a communication strategy that allows for a better understanding of the robustness of analyses across DVs and countries. Maybe, the authors can provide such an overview in a visualization? Ideally, this visualization would allow the reader to better understand where there are robust findings and where there is heterogeneity in the statistical results.**

We appreciate that the number of analyses may be overwhelming for the reader. For ease of readability, we have now included a new table 1 in the discussion on p. X which summarises the results and their implications for our hypothesis testing.

5) It would be good to have descriptives and correlational tables involving the DVs in the Supplementary Materials.

We agree that this would clarify important information for the reader. We have now added descriptives tables for each DV with sample size *N*s, means, and standard deviations for each group in the Supplement. Furthermore, we have included a correlation matrix including all variables in the Supplement.

6) I suggest to use the terms direct and indirect instead of explicit and implicit. The latter terms seem to imply assumptions about the reflectiveness of cognitive processes that might not apply in the way the measures are used.

We appreciate that our original terminology may have been confusing to some readers. Due to these measures being used in previous work, specifically by Roozenbeek and colleagues (2022), we have decided to adopt similar terminology to the authors: our “explicit manipulation discernment” now becomes “technique discernment”, and “explicit detection of (non)-manipulative content” becomes “technique recognition of (non)-manipulative content”. Furthermore, Roozenbeek and colleagues (2022) measured trustworthiness in a similar way to our measure of manipulateness. As a result, we have decided to change “implicit manipulation discernment” to simply “manipulation discernment” and to simply remove the “implicit” from “implicit manipulateness assessments of (non)-manipulative content”.

7) This paper potentially allows for increasing our understanding of the specificity of intervention effects and I believe that this perspective would substantially increase the scientific merit of the paper. In the Supplementary Materials, the authors report analyses on spillover effects of interventions on the identification of manipulative tactics that have not been prepunked. Some of these analyses indicate significant effects in a similar range as the effects on related outcomes. I suggest that the authors include this as an important finding in the paper and discuss the implications. How can these effects be explained? Does this challenge the theoretical assumptions for how the interventions work? Where do we find spillover effects and where not? This question of specificity vs. spillover effects seems to be important to discuss in the paper.

Thank you for raising this important consideration. We have now included references to the cross-protection effects in all results sections where relevant. For example, p. 13 now reads:

“The long scapegoating video (vs. control) also significantly increased manipulateness assessments of the manipulative decontextualization and discrediting content (see Supplement, Section 1).”

Furthermore, we have now added a theoretical discussion for these findings in the general discussion on pp. 33-34:

“Extending Roozenbeek and colleagues’ (2022) findings and replicating other similar work (e.g., Biddlestone et al., 2025; Roozenbeek et al., 2022), we also found evidence of positive

spill-overs or “cross-protection”, wherein prebunking one manipulation tactic improved the detection of another, unrelated tactic. For example, the long decontextualization video significantly increased manipulation discernment of both the scapegoating and discrediting content compared to the control group, and each of the three long videos significantly increased technique recognition of the manipulative content for at least one other tactic (see Supplement, Section 1). Theoretically, this indicates that on top of straightforward educative effects (i.e., teaching participants what a specific tactic is and how to spot it), the inoculation videos tested here may also confer a more generalized resistance schema for discerning manipulative information from reliable content. As a result, we recommend this technique-based inoculation approach over more content-specific interventions such as fact checking due its scalability in reducing misinformation susceptibility across various domains (see also Roozenbeek et al., 2022). Interestingly, although those who indicated having seen any of the videos before reported the highest confidence in detecting manipulation, those who indicated not having seen any of the videos previously were most likely to demonstrate the highest detection and discernment of relevant manipulation (see Supplement, Section 5). This suggests that psychological booster shots of prebunking messages are vital to provide individuals with the skills to detect misinformation that match assessments of their own abilities (see Maertens et al., 2025).”

- 8) **In regard to cross-cultural variation, it seems important to note that the cultural heterogeneity in the selection of nations is limited since they are all European countries.**

We agree that this is an important issue to raise. We have now added this to the discussion on pp. 34-35:

“While some cross-cultural heterogeneity is expected, this mixed evidence coupled with the fact that all countries included in the current sample were European does raise an important question: Are we underestimating the cross-cultural variability in the efficacy of inoculation interventions (see also Harjani et al., 2023)?”

Reviewer 2’s comments

- 1) **The scale of this study, with almost 20000 participants in 12 different nations, is admirable, and I believe the assessment of the effectiveness of misinformation inoculation across different populations is both timely and important.**

Thank you for your kind words about the meaningful contribution of our work.

- 2) **Primarily, I found the argued benefits of inoculation tactics to be overstated based on the current results, and there to be insufficient discussion on both (1) the results that did not support the hypotheses, and (2) when the intervention either primarily or exclusively shifted participants’ response bias to be more conservative rather than improved discernment. Given the current study was run under idealistic conditions (i.e., experimental setting where ratings of information occurred immediately after exposure to the intervention), the size and inconsistency of the effects is, to me, somewhat**

damning. Indeed, even if there is just a small decay in the effectiveness of the intervention after a delay (which research suggests to be likely), it is difficult to see these short interventions being impactful in practice (especially given it is unlikely people would view a 50 second ad in practice). I do believe these findings are worth publishing, but this should be done in a way that more accurately reflects the strength and limits of the evidence.

Thank you for raising this important consideration. Please refer to our new Table 1 on p. X to address a similar point from Reviewer 1. Specifically, this table includes information on how each finding faired in relation to the respective hypotheses.

We have also added a reference to the implication for these small effects in the field on p. 35:

“In sum, despite the promising significant aggregate effects, effect size magnitudes were notably small, warranting caution when expecting these interventions to work in the field (see Roozenbeek et al., 2022). Nevertheless, fieldwork testing the efficacy of these videos on over 1 million viewers garnered generally positive results (see Google Jigsaw, 2022), and similar fieldwork suggests that we can remain confident about the efficacy of prebunking videos for approximately 24 hours after viewing the content (see Roozenbeek et al., 2022).”

Furthermore, we have included an additional discussion of the exploratory sharing intentions hypotheses on p. 39:

“Alongside the reduced technique recognition of non-manipulative content, the long scapegoating video actually increased willingness to share relevant manipulative content, and only the long decontextualization video significantly reduced willingness to share relevant manipulative content. While our hypotheses on sharing intentions were exploratory, these findings have implications for the worsening of information hygiene in online spaces at first glance. However, the increased willingness to share non-manipulative content after watching all videos (long and short) translated into improved sharing discernment across all videos. Furthermore, the effects on sharing intentions appeared to be most pronounced among those who were already likely to share content online, as indicated by the moderation effects by intentions to share the videos within one’s social network (see Supplement, Section 2). Taken together, these findings indicate that the inoculation videos presented here likely motivated participants’ overall engagement with sharing any content online. While this risked higher intentions to share manipulative content, their overall decisions to share reliable content over manipulative content were still improved.”

Please refer to the discussion in its entirety for our references to each finding and what they mean for our hypotheses.

- 3) Relatedly to the first point, in some conditions, the intervention appears to increase confidence in identifying manipulative tactics without improving actual accuracy. This seems like a negative or even potentially harmful outcome, yet it is not acknowledged as such in the discussion. The variability in effects across the three manipulation types and across populations further suggests that the intervention’s impact is, at best, inconsistent. These results seem to challenge the commonly cited generalizability and scalability of inoculation strategies. I recommend that these limitations be more**

prominently addressed in the discussion, including more focus on which populations the intervention is—and is not—effective in.

We would like to draw your attention to the Table 1 summary that we have added. The only instances in which confidence does not match the intervention effectiveness is when confidence is reduced but detection or discernment of manipulation is improved. In other words, people are actually unaware of their improved skills rather than being provided a false sense of confidence.

We agree that deeper discussion about the details of the heterogeneity of effects between surveys is indeed warranted beyond the Supplement. Therefore, we have added more information about the main effects by survey on pp. 34-35 in the discussion:

“Improvements in manipulation discernment were only indicated in Belgium, Germany, Hungary, Italy, and Poland, improvements in technique discernment were only indicated in Bulgaria, Germany, Hungary, Netherlands, Romania, and Spain, and exclusively null effects for sharing discernment were indicated in Lithuania, Netherlands, and Romania (see Supplement, Section 3). While some cross-cultural heterogeneity is expected, this mixed evidence coupled with the fact that all countries included in the current sample were European does raise an important question: Are we underestimating the cross-cultural variability in the efficacy of inoculation interventions (see also Harjani et al., 2023)? Furthermore, our average effect size of $d = 0.08$ for technique discernment did not achieve the average power of .80 to detect an effect size of $d = 0.20$ in our samples, warranting caution when interpreting our findings. In sum, despite the promising significant aggregate effects, effect size magnitudes were notably small, warranting caution when expecting these interventions to work in the field (see Roozenbeek et al., 2022). Nevertheless, fieldwork testing the efficacy of these videos on over 1 million viewers garnered generally positive results (see Google Jigsaw, 2022), and similar fieldwork suggests that we can remain confident about the efficacy of prebunking videos for approximately 24 hours after viewing the content (see Roozenbeek et al., 2022).”

- 4) As with similar studies in this area, I hold concerns about the appropriateness of claims used to assess the effectiveness of the intervention.**
- a) For one, being able to identify a potentially manipulative tactic is not equivalent to being able to discern true and false information, and I find the presentation of these findings as targeting misinformation to be misleading. Moreover, it is unclear to me how the paper is directly targeting election content, which is presented as one of the key aims of the piece. These factors draw into question the appropriateness of the design to target the key research question.**

We would like to draw your attention to our definition of misinformation susceptibility on p. 3:

“Research has demonstrated this strategy to be effective at reducing misinformation susceptibility—that is, in the perceived reliability of false or misleading information, regardless of the intention to deceive (see van der Linden et al., 2023; van der Linden et al., in press).”

Our definition follows the consensus report from the American Psychological Association (van der Linden, 2023). At current, it is our understanding that misinformation is not defined solely by outright factual inaccuracy. Instead, misinformation is false or *misleading* information. Importantly, work conducted by Allen and colleagues (2024) found that unflagged doubts on vaccine safety had 46-fold higher impact on vaccine hesitancy than flagged misinformation (that tended to be outright false; see also van der Linden & Kyrychenko, 2024). Therefore, while the detection of manipulation indeed does not necessary carry implications for detecting factual inaccuracies, detecting factual inaccuracies was not the goal of the current interventions. More importantly, interventions to improve the detection of manipulation are in fact attempts to tackle misinformation given that most consequential misinformation contains manipulation, and are arguably *more* important than interventions teaching people to detect factual inaccuracies or singular myths.

Regarding your concern over election misinformation, we would like to draw your attention to this information included in our abstract:

“These videos targeted three widely used misinformation tactics—scapegoating, decontextualization, and discrediting—which were prevalent across EU nations leading up to the elections.”

As well as this information on pp. 4-5 of our introduction:

“Tackling mis- and disinformation in a rights-preserving manner is an urgent societal challenge (WEF, 2024). To address this, Moonshot (a research-based non-profit tackling online harms), Jigsaw (a unit within Google), and Google developed inoculation videos targeting three manipulation tactics commonly seen during election cycles. The first video covered scapegoating, highlighting how individuals or groups are unfairly blamed for complex issues they are not solely responsible for (Orosz et al., 2024a; Roozenbeek et al., 2022). The second addressed decontextualization, showing how information can be distorted when removed from its original context to present content in a biased or manipulative way (Parihar et al., 2025). The third tackled discrediting, exposing how personal attacks are used to undermine credibility (Orosz et al., 2024b). During a paid media campaign for the June 2024 EU elections, these videos were distributed across YouTube and Meta in Germany, Belgium, France, Italy, and Poland. While Google measured effectiveness via YouTube brand lift surveys (average fieldwork improvement = 1.53%; translates to approximately 9.18% improvement in lab settings; see Mertens et al., 2021; Roozenbeek et al., 2022), it was not possible to evaluate videos created for other European languages, which also had native-language voiceovers. We therefore tested their impact on manipulation discernment in controlled lab experiments across 12 nations, three months after the elections.”

To clarify, the tactics were found to be prevalent during the run-up to the 2024 EU election cycles and were therefore selected as tactics to prebunk. Our reasoning is that since these are prevalent manipulation tactics used during elections, prebunking them is likely to help in the effort to fight election misinformation.

- b) The decontextualized claims rely on the participants imagining being presented with a claim, rather than a claim actually being presented. This creates a confound where for some claims the participant could reasonably imagine the claim to be manipulative or non-manipulative, unrelated to how the authors have classed these claims. For example, a clip of a comedian making**

a fan cry could plausibly happen, and isn't necessarily manipulative, whilst a picture of a line of people with the caption about attending concerts despite cost of living could easily be an image taken out of context. Similarly, certain claims in the 'discrediting' condition (e.g., "researchers work hard to receive funding") could easily be classed as an oversimplification. These issues with the claims draws into question the ability to draw relevant inferences about participants ability to discern manipulative and non-manipulative content.

Thank you for raising this important concern. We fully agree that participants' imagined context may influence their perception of manipulateness, particularly for claims that could plausibly be interpreted in multiple ways. To address this issue empirically, we conducted a series of robustness checks using the manipulateness assessments data of both manipulative and non-manipulative content for each of the three manipulation tactics on participants from the control group alone. These analyses confirmed that participants perceived manipulative items as significantly more manipulative than non-manipulative items across all three manipulation tactics, supporting the validity of our manipulateness classifications even under baseline, uninfluenced conditions. We have included reports of these analyses in footnotes on pp. 13-14:

"As a robustness check, we conducted a paired samples t-test comparing manipulateness assessments of the manipulative vs. non-manipulative scapegoating content only on participants in the control group. The manipulative scapegoating content was perceived as significantly more manipulative, $M = 4.49$, $SD = 1.71$, than the non-manipulative scapegoating content, $M = 3.86$, $SD = 1.50$, $t(3,777) = 18.84$, $p < .001$, $d = 0.31$, validating this as a measure of manipulation discernment."

p. 15:

"As a robustness check, we conducted a paired samples t-test comparing manipulateness assessments of the manipulative vs. non-manipulative decontextualization content only on participants in the control group. The manipulative decontextualization content was perceived as significantly more manipulative, $M = 4.46$, $SD = 1.65$, than the non-manipulative decontextualization content, $M = 3.83$, $SD = 1.52$, $t(3,779) = 20.01$, $p < .001$, $d = 0.33$, validating this as a measure of manipulation discernment."

And again on p. 16:

"As a robustness check, we conducted a paired samples t-test comparing manipulateness assessments of the manipulative vs. non-manipulative discrediting content only on participants in the control group. The manipulative discrediting content was perceived as significantly more manipulative, $M = 4.66$, $SD = 1.89$, than the non-manipulative discrediting content, $M = 3.63$, $SD = 1.53$, $t(3,759) = 25.52$, $p < .001$, $d = 0.42$, validating this as a measure of manipulation discernment."

We have also added a clearer acknowledgment in the discussion of these the important considerations you have raised. This can be found on pp. 44-45:

"Unlike previous research which found a minimal confounding influence of the measurement scale used, item sentiments, and order effects in the piloting stage (see Roozenbeek et al.,

2022), the claims used in our item rating task were not formally pilot tested on perceived manipulateness or other potentially confounding dimensions (e.g., emotional tone, complexity). Although each non-manipulative item was designed to mirror its manipulative counterpart in content and length, it is possible that unintended differences may have influenced participant ratings. To validate the categorization of manipulative and non-manipulative claims, we conducted paired samples *t*-tests on participants' perceived manipulateness ratings. As intended by the authors, manipulative claims were rated as significantly more manipulative than non-manipulative claims in the control group alone. These analyses confirmed the same pattern across all three manipulation tactics: claims in the scapegoating, decontextualization, and discrediting measures that were labelled as manipulative were perceived as significantly more manipulative than non-manipulative claims (all *ps* < .001), supporting the robustness of our assumed categorization under baseline conditions. Furthermore, we ran exploratory multilevel models accounting for individual item variation (see Supplement, Section 6). In all cases, effects remained similar to the main analyses, and the vast majority of variance was accounted for by individual differences between participants, rather than item variation. Nevertheless, we acknowledge that participants' interpretations of claims—especially in the absence of full context—may vary. For example, some may view a celebrity making a fan cry as sincere rather than manipulative, or consider a neutral-sounding statement like “researchers work hard to receive funding” as an oversimplification rather than an instance of discrediting. While our classifications were statistically confirmed in the control group, this interpretive variability may contribute to unexplained variance in discernment scores and reflects broader challenges in operationalizing manipulation in brief, especially in cross-national research.”

- c) **It is also unclear to me how these claims have been matched other than including somewhat similar content. For example, were these claims pilot tested on the important characteristics of interest? This could easily impact the inferences drawn - e.g., if ‘non-manipulative’ and ‘manipulative’ content are starting from very different points on measures of interest (e.g., ratings of manipulateness) I worry that subtraction scores would be impacted by the scale used (indeed, I worry about the information lost by using subtraction scores more generally). I think the reader would also benefit from figures with absolute values (rather than unstandardized beta weights) as this is more intuitive to understand the size of the effects.**

We acknowledge that the manipulative and non-manipulative claims were not formally pilot tested prior to inclusion in the study. Instead, they were constructed to mirror each other closely in topical content, format and length, with the main point of contrast being the presence or absence of a known manipulation technique. While this design choice allowed us to include a broad set of real-world examples across 12 countries, we agree that formal pilot testing could help strengthen the internal validity of such comparisons and have added this as a limitation in the manuscript. This can be found on p. 44:

“Unlike previous research which found a minimal confounding influence of the measurement scale used, item sentiments, and order effects in the piloting stage (see Roozenbeek et al., 2022), the claims used in our item rating task were not formally pilot tested on perceived manipulateness or other potentially confounding dimensions (e.g., emotional tone, complexity). Although each non-manipulative item was designed to mirror its manipulative counterpart in content and length, it is possible that unintended differences may have influenced participant ratings.”

We also note again that despite the lack of pilot testing, our control group analyses confirm that manipulative claims were perceived as significantly more manipulative than non-manipulative ones across all three tactics, providing post hoc support for the validity of our classifications. We also analysed exploratory multilevel models controlling for the individual items as random intercepts. These confirmed that while some variance could be attributed to different items, the vast majority of variance was due to individual differences. References to these analyses can be found on p. 44:

“To validate the categorization of manipulative and non-manipulative claims, we conducted paired samples t-tests on participants’ perceived manipulateness ratings. As intended by the authors, manipulative claims were rated as significantly more manipulative than non-manipulative claims in the control group alone. These analyses confirmed the same pattern across all three manipulation tactics: claims in the scapegoating, decontextualization, and discrediting measures that were labelled as manipulative were perceived as significantly more manipulative than non-manipulative claims (all ps < .001), supporting the robustness of our assumed categorization under baseline conditions. Furthermore, we ran exploratory multilevel models accounting for individual item variation (see Supplement, Section 6).”

We also appreciate your concern regarding potential information loss from the use of subtraction scores. This is why we also included analyses that present manipulateness ratings for manipulative and non-manipulative content separately. This helps ensure that interpretation is not reliant solely on difference values and clarifies whether effects stem from increases in perceived manipulateness, decreases, or both.

We also appreciate your recommendation to include plots of the means for each outcome variable between conditions. We have now added these to Section 7 of the Supplement and referred readers to this section in our analysis plan. Please also note that while we understand your concerns regarding the unstandardised beta forest plots, we included these as they are standard practice when reporting the effects for multilevel models. We have also included standardised Cohen’s *d* coefficients for all effects in-text for the relevant analyses, so we hope that the inclusion of these options is satisfactory for you to fully understand the analyses presented.

Finally, we have added reliability statistics to the relevant variables in the *Measures* subsection of the **Methods** section for further clarity.

- 5) There appears to be some deviation from the pre-registered analysis plan (or at least it is unclear to me as a reader whether the analyses presented are the same as those pre-registered). I would recommend the authors outline any instance where the analyses presented deviate from the pre-registration, or more clearly state the analyses used.**

Thank you for flagging this lack of clarity. We have now included a list of the directional vs. exploratory pre-registered hypotheses before the results section on pp. 9-11:

“Pre-registered directional hypotheses

H1) Watching a relevant prebunking video will significantly increase implicit manipulation discernment (i.e., rating manipulative content as more manipulative and non-manipulative content as less manipulative compared to the control group) for that tactic.

H2) Watching a relevant prebunking video will significantly increase explicit manipulation discernment (i.e., correctly identifying the relevant manipulation tactic for manipulative items and “None of the above” for non-manipulative items more frequently than the control group) for that tactic.

H3) Watching a relevant prebunking video will significantly increase implicit detection of manipulation (i.e., higher manipulateness ratings for manipulative items compared to the control group) for that tactic.

H4) Watching a relevant prebunking video will significantly increase explicit detection of manipulation (i.e., correctly identifying the relevant manipulation tactic for manipulative items more frequently than the control group) for that tactic.

H5) Watching a relevant prebunking video will significantly improve sharing decisions (i.e., lower willingness to share manipulative items and higher willingness to share non-manipulative items compared to the control group) for that tactic.

H6) Watching a relevant prebunking video will significantly decrease willingness to share manipulative content compared to the control group for that tactic.

Pre-registered exploratory research questions

RQ1) Will watching a relevant prebunking video significantly affect implicit detection of non-manipulative content (i.e., lower manipulateness ratings for non-manipulative items compared to the control group) for that tactic?

RQ2) Will watching a relevant prebunking video significantly affect explicit detection of non-manipulative content (i.e., more frequent correct “None of the above” responses for non-manipulative items than the control group) for that tactic?

RQ3) Will watching a relevant prebunking video significantly affect willingness to share non-manipulative content (i.e., higher willingness to share non-manipulative items compared to the control group) for that tactic?

RQ4) Will watching a relevant prebunking video significantly affect confidence in detecting manipulation compared to the control group for that tactic?

RQ5) Cross-protection: Will watching a prebunking video for one tactic improve implicit manipulation discernment, explicit manipulation discernment, and sharing decisions compared to the control group for other, non-prebunked tactics?

RQ6) Skepticism/distrust: Will any prebunking condition increase skepticism (i.e., rating both manipulative and non-manipulative content as more manipulative, or selecting a manipulation tactic for both manipulative and non-manipulative items more frequently than the control group)?

RQ7) Naïveté/gullibility: Will any prebunking condition increase naïveté (i.e., rating both manipulative and non-manipulative content as less manipulative, or selecting “None of the above” for both manipulative and non-manipulative items than the control group)?

RQ8) Will any hypothesized or exploratory effects be moderated by any of the following variables: general manipulation discernment, voting behaviour in the June EU elections, intentions to share videos within one’s social network, digital literacy, socio-demographics (age, gender, education, political ideology), or country-level indices (latitude, education index, industrialisation index, GDP per capita, democratic index)?

Pre-registration deviations

In our pre-registration document, we also planned to explore whether any of the effects were moderated by video length. Furthermore, we collected information on whether participants believed they had seen, not seen, or were uncertain whether they had seen the video in question before. While we were unable to include these variables as moderators because participants in the control group could not provide comparable information, we instead analyzed whether there were any significant differences in our main outcomes variables by

video length and whether participants believed they had seen the videos before. The results for these analyses can be found in the Supplement (Sections 5 and 6)."

- 6) The abstract appears to suggest that the 50 second videos are significantly more effective than the 20 second, however, the relative effectiveness of the different videos does not appear to be directly compared.**

Thank you for noticing this lack of clarity. We have now edited the abstract so that it reads:

"Longer (50s) videos showed more consistent improvements in discernment than shorter (20s) ones, but both improved technique discernment."

Reviewer 3's comments

- 1) Generally, I found the Introduction to be well-written, and I understand the authors are probably working with space constraints. That said, I found certain theoretical pieces to be lacking. In their absence, it was at times difficult to evaluate the novelty and importance of the study. I offer a handful of suggestions here to bolster the theoretical framework.**
 - a) It would have been helpful to contextualize and set up the DVs in the Introduction. I was not quite prepared to interpret the Results. A subsection dedicated to explaining the study outcomes in the Introduction would be very helpful.**

We appreciate the need for more detail on these DVs. Therefore, we have added information contextualizing these in the introduction on p. X:

"To understand how individuals engage with manipulative content online, we examine four key outcome variables that together capture different aspects of misinformation discernment and susceptibility. First, technique discernment refers to individuals' abilities to correctly identify the manipulative techniques (e.g., scapegoating) embedded in content (see also Roozenbeek et al., 2022; Appel et al., 2025). This measure taps into cognitive awareness of the categorization of manipulation tactics and reflects an important component of media literacy—the skill of detecting influence strategies designed to mislead or persuade. Second, manipulation discernment captures participants' ability to judge the relative manipulateness of content, as indicated by higher ratings of manipulative items compared to non-manipulative ones (e.g., Roozenbeek et al., 2022). This measure reflects a more intuitive or implicit evaluation of content credibility. Third, we assess sharing intentions, operationalized as participants' stated willingness to share manipulative versus non-manipulative content. Sharing intentions are a critical behavioral proxy for misinformation spread, directly relevant to real-world consequences of exposure (e.g., Guinote et al., 2025), and even indicative of real-world sharing habits (see Goldberg et al., 2024; Mosleh et al., 2020). Finally, confidence in manipulation detection measures how sure participants are in their ability to tell when content is manipulative. This metacognitive component provides insight into individuals' self-perceived epistemic vigilance—important because overconfidence or underconfidence may influence whether people act on their judgments (e.g., by sharing or fact-checking).

Together, these dependent variables offer a comprehensive picture of how individuals process, evaluate, and respond to manipulative content.”

b) There are a multitude of theoretically relevant moderators. Why did the authors focus on political tolerance and digital literacy relative to other potential moderators? Note, I do not think the authors should have necessarily included other moderators, but I do think they need to explain and justify why these moderators were important to focus on.

We agree that our theoretical justifications for the inclusion of these moderators were lacking in clarity. We have added more detail to this in our introduction on pp. 7-8:

“To understand how individuals engage with manipulative content online, we examine four key outcome variables that together capture different aspects of misinformation discernment and susceptibility. First, technique discernment refers to individuals’ abilities to correctly identify the manipulative techniques (e.g., scapegoating) embedded in content (see also Roozenbeek et al., 2022; Appel et al., 2025). This measure taps into cognitive awareness of the categorization of manipulation tactics and reflects an important component of media literacy—the skill of detecting influence strategies designed to mislead or persuade. Second, manipulation discernment captures participants’ ability to judge the relative manipulateness of content, as indicated by higher ratings of manipulative items compared to non-manipulative ones (e.g., Roozenbeek et al., 2022). This measure reflects a more intuitive or implicit evaluation of content credibility. Third, we assess sharing intentions, operationalized as participants’ stated willingness to share manipulative versus non-manipulative content. Sharing intentions are a critical behavioral proxy for misinformation spread, directly relevant to real-world consequences of exposure (e.g., Guinote et al., 2025), and even indicative of real-world sharing habits (see Goldberg et al., 2024; Mosleh et al., 2020). Finally, confidence in manipulation detection measures how sure participants are in their ability to tell when content is manipulative. This metacognitive component provides insight into individuals’ self-perceived epistemic vigilance—important because overconfidence or underconfidence may influence whether people act on their judgments (e.g., by sharing or fact-checking). Together, these dependent variables offer a comprehensive picture of how individuals process, evaluate, and respond to manipulative content.”

c) Similarly, what did the authors expect for the WEIRDness, demographics, and general manipulation discernment moderation results? What do these analyses tell us? I wasn’t sure how to interpret the moderation results without the necessary background.

While these were exploratory, they were used to explore whether the interventions were broadly effective or whether there were factors requiring attention to improve receptivity. We have added an explanation of this on p. 6:

“To understand whether the prebunking interventions demonstrated broad efficacy across populations, or whether there are certain populations and factors that require attention to improve receptivity to these interventions, we included several moderator variables.”

2. Similarly, I think some crucial elements were missing from the Discussion.

Again, in the absence of theoretical interpretation, it can be difficult to situate the study within the larger literature.

- a) More real estate should be dedicated to walking through the moderation results. What do the smattering of significant results tell us about the constructs and the theoretical framework?**

Thank you for this helpful recommendation. We have now spent more time theoretically interpreting the moderation effects in our discussion on pp. 35-37:

“A number of interesting moderation effects also emerged, particularly with regard to country-level factors. For example, the scapegoating videos appeared to only improve technique discernment of the scapegoating content among participants from more Western or central European nations, and nations with moderate or higher education indices, GDP per capita, and democratic indices. A number of these may be attributed to ceiling effects, such as the particularly high baseline discernment of scapegoating content indicated among participants from more Eastern European nations (see Supplement, Section 2). The conditional effects of the education indices of nations coupled with similar effects for personal educational attainment indicate that higher levels of education at both the country- and individual-level improve participants’ receptivity to these prebunking interventions. Therefore, future research should focus on teaching general educational skills that improve individuals’ critical thinking and provide them with the general ability to effectively appraise and evaluate the veracity of evidence outside of the prebunking context to ensure intervention effectiveness when implemented. In contrast, many of the videos appeared to be effective regardless of participants’ general manipulation discernment abilities, age groups, levels of digital literacy, or voting behavior during the June 2024 EU elections. The lack of conditional effects of these variables on prebunking indicates that initial concerns over the role of these factors potentially dampening the effectiveness of misinformation interventions may be excessive.

At the same time, several variables—particularly political tolerance and political ideology—did appear to interact with the effectiveness of certain videos, suggesting that individuals who are already dispositionally open to alternative viewpoints may be more likely to benefit from prebunking. This aligns with past research on motivated reasoning and ideological asymmetry, in which higher political tolerance and openness are associated with a greater willingness to consider corrective or dissonant information (e.g., Shrout et al., 2022). For example, the scapegoating and discrediting videos were more effective among politically left-leaning or more politically tolerant participants, while effects were weaker or non-significant among centrist or right-leaning groups. This pattern raises important questions about how to reach individuals who may be more ideologically rigid or resistant to interventions, especially since these individuals may also be more vulnerable to manipulative content. The interplay between individual-level and structural moderators also provides insight into the mechanisms through which prebunking works. That both personal educational attainment and national education indices moderated effectiveness suggests that interventions rely not only on individual cognitive capacity but also on broader educational infrastructure. These findings are consistent with dual-process models of reasoning, which highlight the role of both individual motivation and environmental cues in determining whether people engage in effortful evaluation (see Bodle & Burleson, 2015). In societies with stronger education systems or higher average educational attainment, people may be more habituated to critically appraise the intent behind persuasive messages, making them more receptive to prebunking messages.

Similarly, the moderating effects of GDP per capita and democratic indices suggest that the efficacy of these prebunking videos is shaped by broader political and economic conditions. The finding that some videos were more effective in more democratic or wealthier nations may reflect greater baseline media literacy, trust in institutions, or prior exposure to civic or critical thinking education. Conversely, in countries with very high or very low GDP or democracy scores, ceiling or floor effects may limit the observable impact of interventions. This indicates that while prebunking can be adapted across contexts, its success may hinge on tailoring the design or delivery to local structural conditions.

Interestingly, in contrast to some theoretical concerns, general manipulation discernment ability and digital literacy only occasionally moderated effects, and often in directions that favoured already more discerning or digitally competent participants. While these results might suggest that people with stronger baseline abilities are more likely to benefit from prebunking, they also raise concerns about potentially widening informational inequalities—if such interventions are not made accessible and engaging for those with lower literacy or cognitive sophistication, the gap between the most and least resilient audiences may grow. Taken together, the moderation results underscore that although prebunking interventions can be broadly effective, they are not uniformly so. Rather, their impact is conditional on both individual-level characteristics (e.g., education, political tolerance) and contextual factors (e.g., education infrastructure). Recognizing these boundary conditions is essential for designing future interventions that are equitable, scalable, and sensitive to the populations they aim to support.”

b) The authors did not really speak to the fact that participants were significantly more willing to share manipulative discrediting content after watching the short discrediting video relative to control. Discussion and interpretation for this effect are warranted.

We have included further exploration of these findings on pp. 39-41:

“Alongside the reduced technique recognition of non-manipulative content, the long scapegoating video actually increased willingness to share relevant manipulative content, and only the long decontextualization video significantly reduced willingness to share relevant manipulative content. While our hypotheses on sharing intentions were exploratory, these findings have implications for the worsening of information hygiene in online spaces at first glance. However, the increased willingness to share non-manipulative content after watching all videos (long and short) translated into improved sharing discernment across all videos. Furthermore, the effects on sharing intentions appeared to be most pronounced among those who were already likely to share content online, as indicated by the moderation effects by intentions to share the videos within one’s social network (see Supplement, Section 2). Taken together, these findings indicate that the inoculation videos presented here likely motivated participants’ overall engagement with sharing any content online. While this risked higher intentions to share manipulative content, their overall decisions to share reliable content over manipulative content were still improved.

These results speak to a concern in the wider interventions literature: that interventions, while generally helpful, may occasionally lead to adverse or unintended effects—a phenomenon often described as “backfire” (e.g., Hoes et al., 2024; Swire-Thompson et al., 2020). In our study, such backfire effects were observed in several forms, including reduced technique recognition of non-manipulative content, lower confidence in manipulation detection (especially for decontextualization), and increased willingness to share certain manipulative content. While these effects were typically small and context-dependent, they

raise questions about the limits of prebunking interventions and the mechanisms through which they may inadvertently disrupt rather than reinforce accurate discernment. One possibility is that the increased cognitive load imposed by short, theory-rich videos prompts overgeneralization: participants may become hyper-vigilant to signs of manipulation and mistakenly apply learned labels to benign content. Alternatively, introducing manipulative tactics without sufficient scaffolding may reduce perceived self-efficacy or create doubt, especially when techniques are abstract or unfamiliar (as with decontextualization). These dynamics echo broader debates about psychological reactance, miscalibrated confidence, and the risk of cognitive fatigue when interventions are not carefully designed (see Tang, 2025).

Another alternative explanation mirrors early findings in the debunking literature where initial concerns about “backfire” effects turned out to be mostly due to methodological artefacts such as study design issues, item effects, low reliability of certain items, low power, and sample limitations. Larger trials revealed that such concerns about backfire following debunking were relatively rare, inconsistent, and did not replicate well (Wood & Porter, 2019; Swire-Thompson et al., 2020; 2022). In our case, internal consistency for the non-manipulative item set was only modest ($\alpha \approx .58-.70$), meaning that apparent effects on these items should be interpreted cautiously. As Swire-Thompson and colleagues (2020) noted, low-reliability measures can inflate noise and make spurious patterns that resemble backfire effects more likely, particularly when true effects are small. Furthermore, although analysis of technique skepticism indicated that all videos increased skepticism, the effects of the videos on skepticism measured by the continuous manipulateness assessments were all non-significant (see Supplement, Section 9). Future studies should thus focus on disentangling the reasons behind the notable differences between the effects of inoculation on technique recognition and manipulateness assessment measures.

Attention should also be drawn to the positive unintended intervention effects discovered here, such as the cross-protection of prebunking a manipulation tactic, which, in some cases, conferred resistance against other manipulation tactics. The small and inconsistent “backfire” findings reported here—which were generally weaker than the effects supporting our hypotheses—should be investigated using larger samples with high-reliability items. Overall, while we remain cautiously optimistic about the promise of prebunking, our findings do suggest that intervention designers should consider not just what is taught but how it is processed—balancing clarity, brevity, and conceptual differentiation to avoid confusing or demotivating participants.”

c) I think the authors need to speak more in the Discussion to the potential backfire effects of prebunking in the study (lower confidence, more willingness to share manipulative content, reduced detection of non-manipulative content) and how that links to larger debates in the literature about prebunking. What do these results tell us about debates in the literature and how should we think about prebunking moving forward?

We appreciate this helpful suggestion to improve the transparency of our interpretations. We have now included a section in our discussion dedicated to this on pp. 39-41:

“Alongside the reduced technique recognition of non-manipulative content, the long scapegoating video actually increased willingness to share relevant manipulative content, and only the long decontextualization video significantly reduced willingness to share relevant manipulative content. While our hypotheses on sharing intentions were exploratory, these findings have implications for the worsening of information hygiene in online spaces at

first glance. However, the increased willingness to share non-manipulative content after watching all videos (long and short) translated into improved sharing discernment across all videos. Furthermore, the effects on sharing intentions appeared to be most pronounced among those who were already likely to share content online, as indicated by the moderation effects by intentions to share the videos within one's social network (see Supplement, Section 2). Taken together, these findings indicate that the inoculation videos presented here likely motivated participants' overall engagement with sharing any content online. While this risked higher intentions to share manipulative content, their overall decisions to share reliable content over manipulative content were still improved.

These results speak to a concern in the wider interventions literature: that interventions, while generally helpful, may occasionally lead to adverse or unintended effects—a phenomenon often described as “backfire” (e.g., Hoes et al., 2024; Swire-Thompson et al., 2020). In our study, such backfire effects were observed in several forms, including reduced technique recognition of non-manipulative content, lower confidence in manipulation detection (especially for decontextualization), and increased willingness to share certain manipulative content. While these effects were typically small and context-dependent, they raise questions about the limits of prebunking interventions and the mechanisms through which they may inadvertently disrupt rather than reinforce accurate discernment. One possibility is that the increased cognitive load imposed by short, theory-rich videos prompts overgeneralization: participants may become hyper-vigilant to signs of manipulation and mistakenly apply learned labels to benign content. Alternatively, introducing manipulative tactics without sufficient scaffolding may reduce perceived self-efficacy or create doubt, especially when techniques are abstract or unfamiliar (as with decontextualization). These dynamics echo broader debates about psychological reactance, miscalibrated confidence, and the risk of cognitive fatigue when interventions are not carefully designed (see Tang, 2025).

Another alternative explanation mirrors early findings in the debunking literature where initial concerns about “backfire” effects turned out to be mostly due to methodological artefacts such as study design issues, item effects, low reliability of certain items, low power, and sample limitations. Larger trials revealed that such concerns about backfire following debunking were relatively rare, inconsistent, and did not replicate well (Wood & Porter, 2019; Swire-Thompson et al., 2020; 2022). In our case, internal consistency for the non-manipulative item set was only modest ($\alpha \approx .58-.70$), meaning that apparent effects on these items should be interpreted cautiously. As Swire-Thompson and colleagues (2020) noted, low-reliability measures can inflate noise and make spurious patterns that resemble backfire effects more likely, particularly when true effects are small. Furthermore, although analysis of technique skepticism indicated that all videos increased skepticism, the effects of the videos on skepticism measured by the continuous manipulativenness assessments were all non-significant (see Supplement, Section 9). Future studies should thus focus on disentangling the reasons behind the notable differences between the effects of inoculation on technique recognition and manipulativenness assessment measures.

Attention should also be drawn to the positive unintended intervention effects discovered here, such as the cross-protection of prebunking a manipulation tactic, which, in some cases, conferred resistance against other manipulation tactics. The small and inconsistent “backfire” findings reported here—which were generally weaker than the effects supporting our hypotheses—should be investigated using larger samples with high-reliability items. Overall, while we remain cautiously optimistic about the promise of prebunking, our findings do suggest that intervention designers should consider not just what is taught but how it is processed—balancing clarity, brevity, and conceptual differentiation to avoid confusing or demotivating participants.”

- d) **It seems that studying older adults was a primary motivation for the study, yet there was little discussion about what these results tell us about older adults. I think far more is needed in the Discussion regarding what these results show us about discernment in older adults. Similarly, the authors do not really return to the main point of focusing on elections either.**

Thank you for noticing this lack of focus. We have added some more interpretation of what our findings mean for the sample of older adults in our discussion on p. 43:

“The results also offer important insights into the discernment capacities of older adults. Although prior work has found that older individuals tend to be more susceptible to misinformation due to factors such as reduced cognitive flexibility, decreased memory performance, or increased reliance on heuristics (Guess et al., 2019; Brashier & Schacter, 2020), our findings suggest that older adults are nonetheless capable of benefiting from brief prebunking interventions. Across the 12 EU countries sampled, we observed improvements in manipulation and technique discernment, as well as in sharing decisions, despite the fact that all participants were aged 45 or older. This challenges narratives that position older adults as uniformly vulnerable and instead points to the potential of well-designed interventions to support discernment in this age group. While effects were smaller than in comparable studies with younger samples (e.g., Roozenbeek et al., 2022; Orosz et al., 2024), our findings indicate that older populations can still meaningfully engage with and apply the lessons of inoculation interventions, particularly when videos are clear, brief, and accessible.”

Furthermore, we have returned to the context of election misinformation at the end of our discussion on pp. 45-46:

“This study was conducted in the immediate aftermath of the June 2024 EU elections, a period marked by heightened exposure to manipulative political content. The three targeted tactics—scapegoating, decontextualization, and discrediting—were selected precisely because of their widespread use in election campaigns and political misinformation. That the videos improved discernment and sharing decisions for content reflecting these tactics suggests that prebunking may be a particularly timely and relevant tool in democratic contexts, where electoral discourse is vulnerable to manipulation. These findings support the use of prebunking as a scalable intervention during election cycles to help inoculate the public against politically motivated misinformation.”

- e) **I think some of the recommendations (e.g., about particularly vulnerable populations) feel like a stretch considering the heterogeneity of the findings and the inconsistency of the significant moderation results. I encourage the authors to temper their conclusions and/or better justify their conclusions.**

We have now tempered our conclusions on p. 46:

“This work demonstrates that although prebunking videos can improve manipulation discernment and sharing decisions for relevant content among older populations across 12 EU nations, effects are small and heterogeneous. This heterogeneity can be partially explained by conditional factors that reduce the efficacy of this intervention (i.e., less educated individuals in nations with lower education indices). Therefore, we recommend that future research replicates the effectiveness of prebunking videos further afield to non-EU

nations and develops scalable ways to build adjacent skills that improve prebunking receptivity (e.g., through other misinformation-oriented education and skills feedback; see Leder et al., 2024)."

3) Finally, I had concerns about some of the methodological choices and internal consistency statistics.

a) The internal consistency statistics for the manipulativeness ratings for the non-manipulative decontextualization and discrediting statements were rather low. Can the authors speak to other metrics of reliability for these ratings? I worry about the reliability of the difference scores when the reliabilities of these ratings were quite low. Similarly, the reliability of the political tolerance scale was rather low.

Thank you for this important consideration. To address this, we have included reliability statistics for relevant variables (now including discernment measures) in the *Measures* subsection of the *Methods* section for further clarity. We have also run exploratory multilevel models for each analysis accounting for individual item variance as a random intercept. This allowed us to determine whether (1) effects remained consistent when individual item variability was accounted for, and (2) how the individual item variance fared compared to the variance accounted for by individual differences. We have added the details of these analyses to the supplement. For example, here is how we reported one analysis in Section 6, pp. 121-122 of the Supplement:

"The long scapegoating condition (vs. control) had a positive significant effect on manipulation discernment of the scapegoating content, $b = 0.07$, $SE = 0.02$, $t(104105) = 3.99$, $p < .001$, but the short scapegoating video (vs. control) had a non-significant effect, $b = 0.01$, $SE = 0.03$, $t(80250) = 0.30$, $p = .762$. Positive significant effects were also found for the long decontextualization, $b = 0.14$, $SE = 0.02$, $t(104105) = 6.44$, $p < .001$, and discrediting videos (vs. control), $b = 0.09$, $SE = 0.02$, $t(104105) = 4.44$, $p < .001$, but not the short decontextualization, $b = 0.05$, $SE = 0.03$, $t(93760) = 1.49$, $p = .137$, or discrediting videos (vs. control), $b = 0.04$, $SE = 0.03$, $t(93780) = 1.12$, $p = .265$. The conditions had a significant effect on all items, $\chi^2(6) = 52.05$, $p < .001$. The variance attributable to the item level was substantial ($\sigma^2 = 0.10$), indicating meaningful differences between items in average endorsement. However, residual variance remained dominant ($\sigma^2 = 4.10$), indicating that the majority of variance stemmed from individual differences."

In all cases, variance accounted for by individual items was negligible to substantial. However, it was always minimal compared to the individual difference variation, and effects all remained similar to the main analyses.

We have also added a reference to this for readers in our discussion on p. 44:

"Furthermore, we ran exploratory multilevel models accounting for individual item variation (see Supplement, Section 6). In all cases, effects remained similar to the main analyses, and the vast majority of variance was accounted for by individual differences between participants, rather than item variation."

b) Performance on the explicit detection task was rather poor (at least poorer than I might have expected). Can the authors elaborate on whether and to what extent these averages might have impacted the results?

We suggest that this is likely due to the introduction of decoy items in this measure compared to the manipulateness assessments. That is, our presentation of two decoy items in the technique recognition measures is likely to have provided a more explicit opportunity to report a *false positive*, especially for the non-manipulative content.

We have edited a paragraph in our discussion to reflect this point with regards to your point as well as in its original communication on pp. 38-39:

“While the videos tended to improve manipulation discernment through higher manipulateness ratings of manipulative content and manipulateness assessments of non-manipulative content remaining unchanged compared to the control group, technique recognition of non-manipulative content was actually slightly reduced by the prebunking videos compared to the control group. Furthermore, overall performance on this task was relatively poor. Roozenbeek and colleagues (2022) used a similar design to our own, testing the effects of inoculation videos on the technique recognition of relevant manipulative and non-manipulative content. While the videos in Roozenbeek and colleagues’ (2022) work did not alter technique recognition of the relevant non-manipulative content in most cases, the Ad Hominem video did significantly reduce technique recognition of the non-manipulative Ad Hominem content. We suggest that this may be because the multiple choice measures of technique recognition in our studies always included two decoy items (e.g., “Appeal to conflict”; see Supplement, Section 4), potentially introducing confounding uncertainty to participants through their deliberation on concepts that had not been introduced to them in the learning phase. Notable heterogeneity in significant improvements of discernment was also discovered when surveys were analyzed separately (see Supplement, Section 3). This is likely explained by the country-level factors that were found to interact with the effectiveness of the prebunking videos.”

c) General manipulation discernment was calculated as a sum score, implicit manipulation discernment was calculated as a difference score, explicit manipulation discernment was calculated as proportion correct across both types of statements, and sharing discernment was calculated as a difference score. Why did the authors score discernment differently across the different DVs? This not only might affect the results but the interpretation of the results. I would encourage the authors to score discernment in a similar fashion across DVs or better justify why discernment is scored differently across DVs (and how these scoring differences affect the interpretation of what discernment means).

Thank you for noticing this important distinction. Our original intention was to treat categorical binary response measures (i.e., technique discernment and general manipulation discernment ability) as sum scores due to their common usage in this way in the literature (e.g., Maertens et al., 2023; 2024). In contrast, manipulation and sharing discernment as measured by manipulateness assessments and continuous intentions measures tend to be calculated with difference scores in the literature (e.g., Roozenbeek et al., 2022). However, upon re-checking the literature in response to your comment, you are correct to question our measure of technique discernment being calculated as a sum score instead of a difference score. That is, Roozenbeek and colleagues (2022) in fact calculated their technique discernment measures as the proportion of correct non-manipulative responses subtracted from the proportion of correct manipulative responses. To maintain consistency in our findings, we have recalculated our measures of technique discernment in this way to address

your comment. The technique recognition section of our results reflects these differences, and effect sizes are now larger and more consistent than when we used sum scores (although hypothesis-wise, our new findings are basically identical to our original findings).

Importantly however, we have maintained our calculation of the MOCRI measure of general manipulation discernment ability as sum scores instead of difference scores, in line with the original psychometric paper validating the MOCRI scale by Maertens and colleagues (2023). This is because they themselves calculated it in this way and argued that this approach captures a more general recognition ability, as opposed to a more behavioural measure of manipulation discernment. Since we are treating this as a moderator variable, a sum score is more appropriate in this context.

- 4) The power analyses were for effects that were much larger than found for the implicit assessments, and the sensitivity analyses also indicate 80% power to detect effects that were larger than found in the study. I have no problem with the power analysis, as it was theoretically sound and based on prior research. I do think the authors need to speak to the power in their study, however, based on the effect sizes (more than they do currently). Can the authors do a sensitivity analysis based on the identified effect sizes, especially for implicit assessments? What was the likely power to detect those small effects? If the power was low, the authors should speak to that and interpret accordingly.**

Thank you for this helpful recommendation. We have now conducted sensitivity power analysis on all main effects and reported in footnotes in the results section when significant effects did not achieve a minimum of .80 power. Interestingly, insufficient achieved power was only indicated for some of the short videos, with all significant effects for the long videos achieving a minimum of .80 power.

We have also edited a paragraph in our discussion to mention some of the effects that did not achieve sufficient power on p. 33:

“Evidence also indicated that the shorter, 20-second videos also improved technique recognition of the relevant manipulative content (although the effect of the short discrediting video did not achieve a minimum power of .80), sharing decisions for the relevant content (although the effect of the short scapegoating video did not achieve a minimum power of .80), and manipulation discernment for the short discrediting video (although the effect of the short discrediting video on manipulativeness assessments of manipulative discrediting content alone did not achieve a minimum power of .80). These findings replicate previous work demonstrating the effectiveness of shorter (e.g., 15-seconds; Maertens, Roozenbeek et al., 2024) inoculation booster videos at improving the detection of relevant manipulation.”

- 5) I understand this was a secondary analysis, but I am a bit puzzled by the results comparing between groups of participants who had seen, had not seen, or were unsure of whether they had seen the videos before. Why would there be differences between these groups? Was the finding expected? If so, why or why not? I currently do not have an understanding of why there would be these identified differences between the groups.**

Thank you for raising this lack of clarity. This exploratory analysis was intended to help us understand whether previous experiences of prebunking provide participants with an

advantage in the item rating task. We have now added an elaboration and interpretation of these findings in our discussion on pp. 33-34 to further clarify this:

“Interestingly, although those who indicated having seen any of the videos before reported the highest confidence in detecting manipulation, those who indicated not having seen any of the videos previously were most likely to demonstrate the highest detection and discernment of relevant manipulation (see Supplement, Section 5). This suggests that psychological booster shots of prebunking messages are vital to provide individuals with the skills to detect misinformation that match assessments of their own abilities (see Maertens et al., 2025).”

6) The confidence in detecting manipulation subsection of the Results did not have the same organization as prior subsections. Can the authors be consistent and present the results by scapegoating, discrediting, and decontextualization separately? It would go a long way for ease of interpretation.

We have now rearranged this section so that it is separated by the tactic subheadings.

7) I understand the results surrounding higher levels of education, but what does it mean to “teach general educational skills”? What does this recommendation specifically mean? That people should receive more education?

We appreciate that this was a vague description of our recommendation. Please see our edit on p. 35:

“Therefore, future research should focus on teaching general educational skills that improve individuals’ critical thinking and provide them with the general ability to effectively appraise and evaluate the veracity of evidence outside of the prebunking context to ensure intervention effectiveness when implemented.”

Sincerely,
BLINDED